# MIX: A Multi-view Time-Frequency Interactive Explanation Framework for Time Series Classification

**Viet-Hung Tran**[1,*]    **Ngoc Phu Doan**[1,*]    **Zichi Zhang**[1,*]    **Tuan Dung Pham**[1],    **Phi Hung Nguyen**[1],
**Xuan Hoang Nguyen**[2],    **Hans Vandierendonck**[1],    **Ira Assent**[3],    **Thai Son Mai**[1,*]

[1]Queen's University Belfast, UK
{h.tran, h.vandierendonck, thaison.mai}@qub.ac.uk
[2]Institut Polytechnique de Paris, France
{xuan-hoang.nguyen}@ip-paris.fr
[3]Aarhus University, Denmark
{ira}@cs.au.dk

## Abstract

Deep learning models for time series classification (TSC) have achieved impressive performance, but explaining their decisions remains a significant challenge. Existing post-hoc explanation methods typically operate solely in the time domain and from a single-view perspective, limiting both faithfulness and robustness. In this work, we propose *MIX* (*M*ulti-view Time-Frequency *I*nteractive E*X*planation Framework), a novel framework that helps to explain deep learning models in a multi-view setting by leveraging multi-resolution, time-frequency views constructed using the Haar Discrete Wavelet Transform (DWT). MIX introduces an *interactive* cross-view refinement scheme, where explanation's information from one view is propagated across views to enhance overall interpretability. To align with user-preferred perspectives, we propose a greedy selection strategy that traverses the multi-view space to identify the most informative features. Additionally, we present OSIGV, a user-aligned segment-level attribution mechanism based on overlapping windows for each view, and introduce keystone-first IG, a method that refines explanations in each view using additional information from another view. Extensive experiments across multiple TSC benchmarks and model architectures demonstrate that MIX significantly outperforms state-of-the-art (SOTA) methods in terms of explanation faithfulness and robustness.

## 1   Introduction

Time series classification (TSC) is an important task in many domains, such as healthcare [46], finance [44], and supply chain [38]. In recent years, deep learning (DL) models have become increasingly popular in TSC thanks to their ability to capture complex temporal patterns and achieve state-of-the-art (SOTA) performance on a wide range of datasets [59]. However, DL models are considered as black boxes due to their lack of transparency. This makes it difficult to understand or trust their predictions, which is especially critical in sensitive applications such as medical and financial, where interpretability, accountability, and error analysis are essential.

To mitigate this issue, explainable AI (XAI) has emerged as an important direction for improving the interpretability of DL models. XAI methods help users to understand model decisions, which in turn help to increase trust, supports accountability, and enables model debugging. In the context

---

[*]Equal Contribution.

39th Conference on Neural Information Processing Systems (NeurIPS 2025).

of TSC, explanation methods can be broadly divided into ante-hoc and post-hoc categories. Ante-hoc methods aim to make models interpretable by design, for example, decision trees, attention mechanisms [32, 31, 69, 9, 46], or shapelet-based methods [92, 47, 22, 61, 24, 91]. Post-hoc methods, on the other hand, provide explanations after the model is trained and are usually agnostic to the model's internal structure. Since most high-performing DL models are not inherently interpretable, post-hoc XAI has become the common explanation approach for DL models in TSC.

According to the type of explanation output, most methods provide either time point-based explanations [36, 86, 12, 17, 49, 48, 66, 11, 4, 16], which give importance score to individual time points or subsequences-based explanations, which focuses on sub-regions of the time series [8, 33, 27]. Time point-based explanations are more common, especially when models operate directly on raw time series. However, assigning importance scores to each time point often results in fragmented, hard-to-interpret explanations and fails to capture the temporal structure or meaningful patterns in the data. Segment-based methods, which assign scores to segments of time series, have recently gained attention as they provide more meaningful explanations for human [63, 29, 76, 83].

Despite these recent efforts, most existing methods, including both time point-based and subsequence-based approaches, focus only on the time domain. They overlook another important aspect of time series: frequency. Addressing this gap, SpectralX [10] introduces an XAI framework that operates in the time-frequency space, using Short-Time Fourier Transform (STFT) to better reflect the underlying characteristics of time series. While SpectralX represents a step forward, it has two key limitations: it uses a fixed STFT setup, which lacks adaptability across different time series; and it fails to map the frequency-based insights back to the time domain, where users typically interpret results. In addition, the SpectralX deletion/insertion/combined attribution mechanism also has shortcomings of stability with randomness, and false negatives in attribution indicated in our detailed analysis presented in Section C of the Appendices.

More broadly, a key limitation across current post-hoc TSC explanation methods is that they rely on a single view of the time series in either time domain or time-frequency domain. This highlights a research gap: the lack of multi-view explanation frameworks that integrate different perspectives of input to provide a more faithful and robust understanding of DL models.

To address the multi-view research gap and fixed setup of SpectralX, we present a new framework, called **M**ulti-view Time-Frequency **I**nteractive E**X**planation (MIX). MIX aims to improve the interpretability of deep learning models for time series classification (TSC) by explaining multiple time-frequency views of the input through Haar wavelet transforms at various resolutions. Each view captures distinct temporal patterns, enabling a richer and more faithful understanding of model behavior. In addition, we propose an interactive mechanism called *cross-view refinement*, which leverages the explanation from one view to improve those of others, enhancing the overall faithfulness of the explanations. To the best of our knowledge, this is the first approach for using multiple perspectives in a view interactive way in post-hoc TSC explanations. Finally, we incorporate a greedy search strategy to identify and map the most important features across views back to the time domain. This not only improves explanation quality but also offers users valuable insights into the granularity of important segments contributing to the model's prediction.

**Summarization.** In this paper, our contributions are summarized as follows:

First, we propose a new perspective for TSC XAI that explains models from multiple views of time series in both time and frequency domains. Our approach MIX not only generates explanations across different time-frequency views, but also applies cross-view refinement to enhance each explanation and performs a greedy feature selection across views to extract the most important patterns in the view, that is most relevant to the users.

Second, we address key limitations of SpectralX's attribution mechanism by introducing a new method based on Integrated Gradients (IG) applied to overlapping segments, defined by the window and overlap size. This approach yields more faithful and human-aligned explanations by capturing temporal relationship more effectively. Furthermore, we extend this mechanism to a variant called *Keystone-first*, inspired by the keystone species concept in ecology[2]. This adaptation also provides a new way to the refinement of explanations in one view using informative segments from another view, helping cross-view refinement.

---

[2]Keystone species is a species that has a disproportionately large effect on its natural environment.

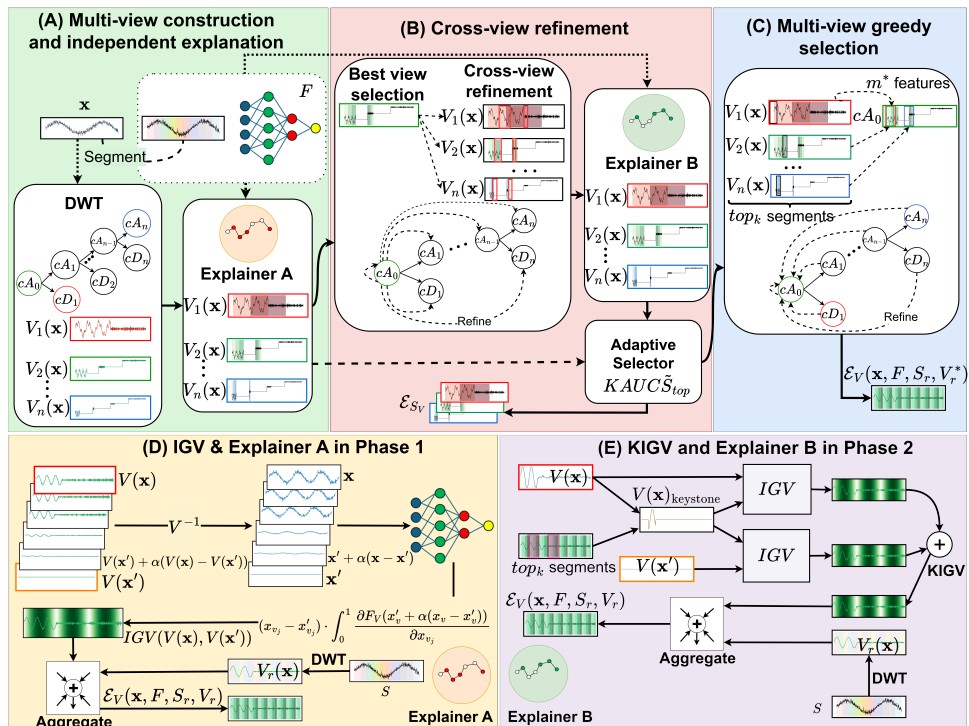

Figure 1: Overview of the MIX framework with three phases. (A) *Multi-view Construction and Independent Explanation* described in Sec 2.5: views $V_r$ are constructed via Haar DWT, then explained independently using IGV and OSIGV. (B) *Cross-view Refinement* described in Sec 2.5: the best view $V_q$ is selected using $\text{KAUC}\tilde{S}_{top}$, then refined using KIGV and OSIGV guided by top-$h$ segments. (C) *Multi-view Greedy Selection* described in Sec 2.5: MIX traverses all views to select key features and maps them to the user-preferred view. Phase 3 is practical for selecting top features directly. (D) Attribution mechanism in Phase 1: IGV is applied to each view, and scores are aggregated into overlapping segments via OSIGV (see Section 2.3). (E) Attribution mechanism in Phase 2: Keystone-first IG for view (KIGV) is used to prioritize keystone features before generate importance score to others (see Section 2.3), then apply OSIGV again to overlapping segments. Refer Section A and Table 3 for meaning of symbols.

Third, we propose a adaptive selector with a novel criterion called Keystone $AUC\tilde{S}_{top}$ ($KAUC\tilde{S}_{top}$), inspired by recent work on attribution evaluation for TSC XAI [84]. The adaptive selector makes the decision whether explanations should be updated after cross-view refinements.

Fourth, we conduct extensive experiments on 11 datasets with 3 deep learning architectures including ResNet-34, BiLSTM, Transformer and compare against 3 state-of-the-art (SOTA) TSC XAI methods including LIMESegment, InteDisUX, SpectralX to demonstrate the performance of MIX.

The Table 1 summarises key differences between MIX and our closest work, SpectralX. In summary, MIX is different to SpectralX in 3 fundamental aspects: (i) hierarchical multi-view explanations, (ii) interactive explanation refinement scheme among views, (iii) novel attribution mechanism.

Table 1: Comparison between SpectralX and MIX.

| Aspect | SpectralX | MIX |
|---|---|---|
| Multi-views | No | Yes |
| Time-frequency transform | STFT | DWT |
| View interaction | No | Yes |
| Attribution mechanism | Insertion/Deletion | OSIGV |

## 2 Multi-view Interactive Explanation

In this section, we describe the proposed MIX framework, as illustrated in Fig. 1, which consists of three phases: *Phase 1: Multi-view Construction and Independent Explanation*, *Phase 2: Cross-view Refinement*, and *Phase 3: Multi-view Greedy Selection*. We begin by clarifying the problem definition

and introducing the concept of multi-view, along with our formulation of the novel multi-view explanation setting. We then define the interpretable representation as the *input* and the explanation as the *output* of MIX. For the internal mechanisms, we define the attribution methods used in each phase: IGV, OSIG, and OSIGV (based on Integrated Gradients) for Phase 1, and KIGV for Phase 2. In addition, we introduce a novel selection criterion, $\mathrm{KAUC}\tilde{S}_{top}$, adapted from the TSC explanation evaluation framework in [84], which is used to select the best view and refine explanations in Phase 2. Finally, we describe the overall algorithmic flow of the MIX framework in addition with description of Phase 3 for practical setups.

## 2.1 Problem Definition

**Time series.** A time series $\mathbf{x} = \{x_1, x_2, \ldots, x_T\} \in \mathbb{R}^{T \times d}$ is an ordered sequence of $T$ real-valued vectors, where each $x_t \in \mathbb{R}^d$ represents the observation at time step $t$. Here, $T$ denotes the total number of time steps, and $d$ is the number of channels. $\mathbf{x}$ is referred to as *univariate* when $d = 1$ and as *multivariate* when $d > 1$. In this paper, we investigate on univariate time series as its popularity in time series explanation. The current SOTA TSC explanation methods are mainly focusing on this type of time series [76, 83, 10]. We drop $d$ out for univariate time series for simplicity.

**Time series classification and explanation.** We denote a TSC dataset is an annotated dataset $D = \{(\mathbf{x}_i, y_i)\}_{i=1}^N$, where $\mathbf{x}_i \in \mathbb{R}^T$ is a univariate time series and $y_i \in \mathbb{N}$ is the corresponding class. The objective of TSC is to learn a DL model $F : \mathbb{R}^T \to \mathbb{N}$ that assigns each $\mathbf{x}_i \in X$ to its corresponding label $y_i$. Our target is to explain the pretrained model $F$ for an instance $\mathbf{x}$ to answer *'Why does F predict x as $F(\mathbf{x})$?'*. As a TSC post-hoc explanation, our approach seeks which part of $\mathbf{x}$ significantly affects the model's prediction outcomes.

**Segment-based explanation for time series.** Different from *Time Points-based Attribution Mechanism*, which assigns an importance score to each time series step, *segment-based attribution method* maintains a set of segments with their scores as the interpretable representation. By considering the time-dependence characteristic of time series, this method makes the explanation more naturally aligned with human perception. Given a time series $\mathbf{x}$, a *segment* is defined as $se_{t_{\mathrm{start}}}^{t_{\mathrm{end}}}(x) = \{x_{t_{\mathrm{start}}}, x_{t_{\mathrm{start}}+1}, \ldots, x_{t_{\mathrm{end}}}\}$ where $t_{\mathrm{start}}$ and $t_{\mathrm{end}}$ are starting time and ending time of the segment and $1 \leq t_{\mathrm{start}} < t_{\mathrm{end}} \leq T$. Let $S = \{se^1, se^2, \ldots, se^M\}$ be a set of segment functions, where each segment $se^i = se_{t_{\mathrm{start}}}^{t_{\mathrm{end}}}$. Our objective is to build an explainer $\mathcal{E}$ that gives a score for each segment $se^i \in S$: $\mathcal{E}(\mathbf{x}, F, S) = \{s_{se^i} | se^i \in S\}$. Here, $|s_{se^i}| > |s_{se^j}|$ indicates that $se^i$ is more important than $se^j$ in making a prediction for the instance $\mathbf{x}$.

**Multi-view Explanation Problem.** The key idea of our explanation method is built upon the concept of multi-view where each view could refine the results of other views and vice versa. Let $V : \mathbb{R}^d \to \mathbb{R}^{d'}$ be a view of $\mathbf{x}$ where it transforms $\mathbf{x}$ to a space of $d'$ dimensions. If the view is invertible, it has a corresponding inverse $V^{-1} : \mathbb{R}^{d'} \to \mathbb{R}^d$ such that $V^{-1}(V(\mathbf{x})) = \mathbf{x}$. Let $S_V$ denote a set of views. Our *multi-view explanation problem* can be defined as to establish a *view-specific explanations* $\mathcal{E}_V(\mathbf{x}, F, S, V)$. Let $\mathcal{E}_V(\mathbf{x}, F, S, V) = \{s_{se^i} \mid se^i \in S\}$ be a set of scores over the segment set $S$ in the view $V$. In particular, $\mathcal{E}_V$ denotes the explanation for the specific view $V$ which is built not only from this view itself but also from the refinement of other views. Finally, we get a set of explanations for the model $F$ as $\{\mathcal{E}_V(\mathbf{x}, F, S, V) \mid V \in S_V\}$.

## 2.2 MIX Framework's Description

**Time-Frequency View Definition.** Recent TSC explanation methods focus on the time domain, using time points or segment-based attribution. They often overlook the frequency domain, which might contain crucial information in many time series. SpectralX addresses this by masking the attribution in the time-frequency domain via Short-Time Fourier Transform (STFT), but its fixed time-frame setup limits adaptability across different time-frequency patterns. To address these limitations, we propose a more flexible and interpretable approach using the Haar Discrete Wavelet Transform (Haar DWT) for time series [7], a widely used signal analysis tool. Wavelet transforms enable multi-resolution decomposition, capturing both time-localized and frequency patterns. In this work, we define views using the multi-resolution Haar DWT. Following standard notation, we denote the DWT outputs as $cA$ (approximation coefficients) and $cD$ (detail coefficients), representing low- and high-frequency components, respectively (see Fig. 1 (A) in DWT part). For multi-level

decomposition, $cA_r$ and $cD_r$ refer to the approximation and detail coefficients at level $r$. We formally define our view by DWT. Given a time series $\mathbf{x}$, the view at level $r$, denoted $V_r$, is defined as: $V_r(\mathbf{x}) = [cA_r, cD_r, cD_{r-1}, \ldots, cD_1]$. In addition, each view $V_r(\mathbf{x})$ can be inverted to reconstruct the original time series $\mathbf{x}$.

**Segments-based explanation in MIX.** We define the interpretable representation used in each view as input to the explanation mechanism. As described above, the view at level $r$ is represented as $[cA_r, cD_r, cD_{r-1}, \ldots, cD_1]$, which can be inverted to reconstruct the original time series $cA_0$. At each level, we focus on explaining on $cA_r$ and $cD_r$, since lower-level detail coefficients are handled in their respective views. With TSC explanation attribution approach, we choose segments-based explanation to conduct as it is more aligned with human perception more than time points based [83]. Given a time series $\mathbf{x} = [x_1, \ldots, x_T]$, window size $w$, and step size $\delta$, the $i$th overlapping segment is: $se^i = se_{t_{\text{start}}^i}^{t_{\text{end}}^i}$, where $t_{\text{start}}^i = \min(1 + i\delta, T)$, $t_{\text{end},i}^i = \min(1 + i\delta + w, T)$. The full set is denoted as $OS_{\mathbf{x}}$. For each $cA_r$ and $cD_r$, the structure of them is similar as a time series, then we generate overlapping segments denoted $OS_{cA_r} = \{cA_r^1, cA_r^2, \ldots, cA_r^N\}$ and $OS_{cD_r} = \{cD_r^1, cD_r^2, \ldots, cD_r^N\}$, and use them as the interpretable representation for explanation. We choose overlapping segments so that explanation can avoid missing some regions between separate segments as in InteDisUX [83]. The window and step sizes are adjusted by scale: $w_r = \min(w \cdot 2^{-r}, 1)$, $\delta_r = \min(\delta \cdot 2^{-r}, 1)$, where $w$ and $\delta$ are sizes defined on the input $\mathbf{x} = cA_0$.

**Definition 2.1** (MIX setup). Given a time series $\mathbf{x}$, deep learning model $F$, a set of views from Haar DWT $S_V = \{V_r \mid r \in [0, m]\}$, where $m$ is the maximum wavelet level, and $w$ and $\delta$ to generate overlapping segments for each $cA$ and $cD$ as input, MIX produces output as a set of explanations with segment set $S_r = OS_{cA_r} \cup OS_{cD_r} \cup \ldots \cup OS_{cD_1}$; $\mathcal{E}_{S_V} = \{\mathcal{E}_V(\mathbf{x}, F, S_r, V_r) \mid V_r \in S_V\}$, where each view's explanation is defined based on segments-based explanation as:

$$\mathcal{E}_V(\mathbf{x}, F, S_r, V_r) = \{s_{se^i} \mid se^i \in OS_{cA_r} \cup OS_{cD_r} \cup \ldots \cup OS_{cD_1}\}$$

## 2.3 Attribution Mechanism for MIX

To get output from input as defined in Def. 2.1, we apply attribution mechanism on overlapping segment. Motivated by limitations in SpectralX's attribution approach analyzed in Sec C of Appendices, we propose a new mechanism, OSIG, based on Integrated Gradients (IG), which satisfies important axioms such as sensitivity and implementation invariance.

**Definition 2.2** (Integrated Gradients (IG) [80]). Given input $\mathbf{x}$, baseline $\mathbf{x}'$, and model $F$, the IG along the $j$th dimension is: $\text{IG}_j(\mathbf{x}) = (x_j - x_j') \cdot \int_0^1 \frac{\partial F(\mathbf{x}' + \alpha(\mathbf{x} - \mathbf{x}'))}{\partial x_j} \, d\alpha$.

Although IG is theoretically sound, it operates at the time point level, which hinders interpretability in TSC. InteDisUX [83] adapts IG for segment-level attribution (SIG), but uses non-overlapping segments, potentially missing important transitional regions. We address this with overlapping segments define in Sec. 2.1.

**Definition 2.3** (Overlapping Segment-level Integrated Gradients (OSIG)). Given segments $\{se^i\}$, OSIG aggregates IG over time points in each segment: $\text{OSIG}(se^i) = \sum_{j=t_{\text{start}}^i}^{t_{\text{end}}^i} \text{IG}_j(\mathbf{x})$.

**Definition 2.4** (IGV and OSIGV). For view $V(\mathbf{x})$, if $V$ and $V^{-1}$ are differentiable, we define IG in view space as:

$$\text{IGV}_j(V(\mathbf{x}), V(\mathbf{x}')) = (V(x)_j - V(x)_j') \cdot \int_0^1 \frac{\partial F(V^{-1}(V(\mathbf{x}') + \alpha(V(\mathbf{x}) - V(\mathbf{x}'))))}{\partial V(x)_j} \, d\alpha$$

To simplify the Def. 2.4, we call the new function with input $x_V = V(x)$ or $x_V' = V(x')$ is $F_V = F \circ V^{-1}$, then Def. 2.4 become:

$$IGV_j(x_v, x_v') = (x_{v_j} - x_{v_j}') \cdot \int_0^1 \frac{\partial F_V(x_v' + \alpha(x_v - x_v'))}{\partial x_{v_j}}$$

To use IGV on segments, we compute OSIG as: $\text{OSIGV}(se^i) = \sum_{j=t_{\text{start}}^i}^{t_{\text{end}}^i} \text{IGV}_j(V(\mathbf{x}))$, where $se_i$ lies on the transformed representation $\{cA_r, cD_r\}$. In MIX, for each view $V_r$, we apply OSIG to overlapping segments of $cA_r$ and $cD_r$, while setting attribution for other $cD_m$ (with $m < r$) to zero.

**Cross-view refinement by attribution mechanism.** While we can directly apply the attribution mechanism OSIGV to obtain the final explanation output $\mathcal{E}_{S_V}$, our goal is to identify the most faithful explanation of the model's decision. To this end, we introduce a refinement scheme called *cross-view refinement*, which leverages attribution information across views. Although each individual explanation may be imperfect, it often highlights features that is strongly important to the model's prediction. These features can be seen as "keystone" features that represent to the core characteristics of the input. Inspired by the ecological concept of keystone species, we define *keystone features* in time series as those with a large impact on model decisions. Analogous to ecological studies that prioritize keystone species, our method first approximates such features using an attribution map, then uses this information to refine the explanation across views. To approximate keystone features, we use the best view $V_r$ choosing by evaluate attribution map of each view by a novel criterion in the next section. Moreover, we choose top-$h$ segments of view $V_r$ with highest score in $\mathcal{E}_V(V_r(\mathbf{x}), F)$. Then, MIX map their positions from view $r$ to view $p$ using the properties of the Haar DWT. Let $cA_r = [x_1, \ldots, x_{T_r}]$ and $cD_r = [x_1, \ldots, x_{T_r}]$. A time step $t_r$ in level $r$ is mapped to level $p$ as: $t_p = \min(\lfloor t_r \cdot 2^{p-r} \rfloor, T_p)$., where $T_p$ is length of $cA_p$ or $cD_p$. For each segment, we apply this mapping to both the start and end positions to locate corresponding segments in $cA$ and $cD$ of the target view. To incorporate this transferred information, we introduce a new attribution method called *Keystone-first Integrated Gradients for view* (KIGV). Inspired by the ecological role of keystone species, KIGV first attributes importance to the top-$h$ keystone segments and then extends attribution to the rest of the view. Since exact keystone features are unknown, we approximate them using the top-$h$ segments from the best view selected using a criterion introduced in the next section.

**Definition 2.5** (Keystone-first Integrated Gradients)**.** Given a view $V(\mathbf{x})$ and top segment positions topV $\subseteq \{1, \ldots, T_r\}$, define a binary mask $\mathbf{mask} \in \{0, 1\}^{T_r}$, where $m_l = 1$ if $l \in$ topV, and 0 otherwise. The masked input is: $V(\mathbf{x})_{\text{keystone}} = \mathbf{mask} \cdot V(\mathbf{x}) + (1 - \mathbf{mask}) \cdot V(\mathbf{x}')$. The KIGV attribution is:

$$\text{KIGV}_i(V(\mathbf{x}), V(\mathbf{x}')) := \text{IGV}_i(V(\mathbf{x})_{\text{keystone}}, V(\mathbf{x}')) + \text{IGV}_i(V(\mathbf{x}), V(\mathbf{x})_{\text{keystone}}).$$

Finally, to refine view $V_m$, we compute KIGV using the transferred segments from $V_r$, and then apply OSIGV to produce a refined explanation guided by cross-view information.

## 2.4 Selection Criterion

In multi-view settings, selecting the most meaningful explanation requires a principled criterion. While InteDisUX uses a heuristic based on discrimination and faithfulness gain, it is not aligned with recent evaluation standards [84]. To address this and support user evaluation preferences, we propose a general, user-adaptable selection method using selection scores based on [84].

First, we define a *top-$k$-quantile-masked sample*, denoted $\tilde{\mathbf{x}}_k^{\text{top}}$, as a perturbed version of input $\mathbf{x}$ where the top-$k$ most important features (based on the absolute attribution scores) are replaced. Features are selected by exceeding the $(1 - k)$-quantile threshold. Unlike [84], which uses random noise, we replace values with the baseline from OSIG, which is the mean value over time series training dataset to maintain stability. The Top-$k$-Quantile strategy typically applies to explanation with scores for each feature, but segment-based attribution provides scores only at the segment level. Given a view $V_r(\mathbf{x})$, to assign scores to individual features in each view $V_r(\mathbf{x})$, we use the maximum absolute score among all segments containing that point: $s_j = \max \left\{ |s_{se^i}| \mid V_r(x)_j \in se^i \right\}$.

To estimate explanation quality, we define the normalized degradation score as the drop in confidence when input $\mathbf{x}$ is perturbed into $\tilde{\mathbf{x}}$: $\tilde{S}(\mathbf{x}) = \frac{S(\mathbf{x}) - S(\tilde{\mathbf{x}})}{S(\mathbf{x})}$, where $S(\mathbf{x})$ is the model's confidence for the predicted class. By progressively corrupting the top- or bottom-$k$ features, we construct $\tilde{S}$–$\tilde{T}$ curves, where $\tilde{T} = \frac{\bar{T}}{T}$ is the fraction of corrupted time steps. The area under this curve quantifies overall degradation: $AUC\tilde{S} = \int_0^1 \tilde{S} \, d\tilde{T}$. Specifically, $AUC\tilde{S}_{\text{top}}$ evaluates degradation from top-$k$-quantile-masked samples with $k \in [0, 1]$. A higher $AUC\tilde{S}_{\text{top}}$ indicates a more faithful explanation. In fact, users often care most about the top influential or "keystone" features. To capture this preference, we propose the following focused selection criterion:

**Definition 2.6** (Keystone $AUC\tilde{S}_{top}$)**.** Keystone $AUC\tilde{S}_{top}$ ($KAUC\tilde{S}_{top}$) is a version of $AUC\tilde{S}_{\text{top}}$, computed over $k \in [0, \kappa)$, where $\kappa \in (0, 1)$ specifies the top fraction of features to prioritize. We denote the function as: $KAUC\tilde{S}_{top}(F, V_r(x), \mathcal{E})$, where $\mathcal{E}$ is explanation on $V_r$.

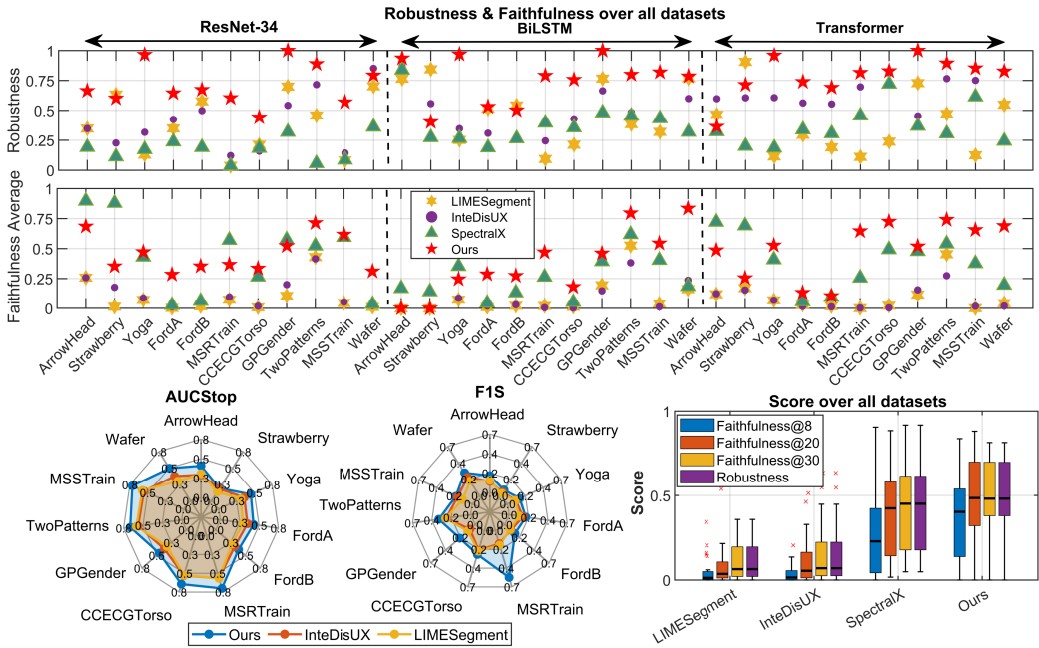

Figure 2: **Comprehensive comparison of our method (MIX) against state-of-the-art baselines.**
**(Top)** Scatter plots show robustness and averaged faithfulness scores across 11 datasets and 3 model architectures. Our method (red star) consistently outperforms others. Quantitatively, MIX ranks first in **27/33** robustness and **24/33** faithfulness evaluations. **(Bottom-left)** Radar plots for $AUCStop$ and $F1S$ metrics further demonstrate the superiority of MIX. **(Bottom-right)** Box plots summarizing the distribution of scores over all datasets confirm that MIX achieves higher and more stable performance.

## 2.5 Multi-view Interactive Explanation

Overall, our algorithm can be divided to 3 phases: Phase A, B, and C as shown in Figure 1.

**Phase 1: Multi-view Construction and Independent Explanation.** Given a time series $\mathbf{x}$, window size $w$, step size $\delta$, and a model $F$, MIX first applies the Haar Discrete Wavelet Transform (DWT) to generate multi-resolution views. Each view $V_r$ corresponds to the $r$-th decomposition level, with $V_0(\mathbf{x}) = cA_0 = \mathbf{x}$, and general form defined in Sec 2.1. In the second step, for each view $V_r$, we generate an explanation by applying the OSIGV attribution mechanism (Def. 2.4). This results explanations: $\mathcal{E}_{S_V} = \{\mathcal{E}_V(\mathbf{x}, F, S_r, V_r) \mid V_r \in S_V\}$.

**Phase 2: Cross-view Refinement.** In the first step, MIX estimate the explanation quality of each view using the $\mathrm{KAUC}\tilde{S}_{top}$ score using Def. 2.6, and then select the best view as following: $V^* = \arg\max_{V_r} \mathrm{KAUC}\tilde{S}_{top}(F, V_r(\mathbf{x}), \mathcal{E}(V_r(\mathbf{x}), F))$. Starting from the selected best view $V^*$ at level $q$ (identified in Phase 1), we first extract the top-$h$ most important segments, denoted $\mathrm{topS}(q, h) \subset OS_{cA_q} \cup OS_{cD_q}$.

In the second step, these top segments are then mapped across all other views using the cross-level mapping defined in Sec. 2.3, producing $\mathrm{topS}(r, h)$ for each level $r$. In the third step, guided by these mapped segments, we apply the keystone-first IG from Def. 2.5 to compute a new attribution for time points then applying OSIG for each view. Finally, we propose adaptive selector for MIX using the $KAUC\tilde{S}_{top}$ selection criterion (Def. 2.6) to choose between the refined explanation and the original explanation from Phase 1, for each view. This results in an updated explanation set, enhanced through cross-view interaction.

**Phase 3: Multi-view Greedy Selection.** After Phase 2, MIX has $\mathcal{E}_V(V_r(\mathbf{x}), F)$ for each view $V_r$. When a target view is required (typically $cA_0$), we aggregate key segments from all levels into it. To do so, we use a greedy strategy that iteratively selects the segment with the highest score from all levels and maps them to the target view, continuing until the desired number of time points is reached.

## 2.6 MIX for Multivariate Time Series.

The MIX framework naturally extends to multivariate time series, as its core components are designed to handle multi-channel inputs. Concretely, in Phase 1, the Discrete Wavelet Transform (DWT) is applied to each channel independently, after which overlapping segments are generated on a per-channel basis. Our attribution methods, including Integrated Gradients (IG) and OSIGV, are then computed directly on the multivariate inputs. The refinement mechanisms in Phase 2 also generalize directly: the top-h segments are selected from the entire pool of segments across all channels based on their scores, and the "KAUC$\tilde{S}_{top}$" criterion is calculated by removing $k$-quantile features based on importance scores aggregated across all channels . For experimental comparisons, we adapted baselines such as InteDisUX and SpectralX by converting the multivariate data into a univariate time series via channel concatenation.

# 3 Experiments

## 3.1 Experimental Setup

**Dataset.** To evaluate our method against state-of-the-art approaches, we first use a synthetic dataset with predefined explanation ground truth. For all, $cA_1$ is the concatenation of three sine waves, and for label 0, $cD_1$ has a constant value of 10 from position 50 to 100. For label 1, $cD_1$ has value 20 in the same region. This corresponds to time steps 100–200 in $cA_0$, reconstructed from $\{cA_1, cD_1\}$. We then apply our explanation method on the time series **x**, obtained via inverse DWT and add random noise to the raw time series. Basically, samples from two label will be different of details in time steps from 100 to 200, noise added in final step will make explanation task more challenging. To demonstrate generalizability on real-world data, we test on 11 UCR datasets: ArrowHead, Strawberry, Yoga, FordA, FordB, MixedShapesRegularTrain (MSRTrain), CinCECGTorso (CCECGTorso), GunPointMaleVersusFemale (GPGender), TwoPatterns, MixedShapesSmallTrain (MSSTrain), and Wafer. In addition, to demonstrate the effectiveness of our explanation method in a real-world scenario, we conduct experiments on the MIT-BIH dataset [66], which includes expert-annotated explanations provided by cardiologists. Full details about the annotation process are described in the TimeX paper [66]. We also conduct our straightforward adaptation on UEA Basic Motion dataset. This dataset's results are in the Appendices Sec. F.1. Full details of dataset are provided in the Appendices, Sec. E.1.

**DL Architecture.** We conduct experiments on three deep learning architectures, ResNet34, BiLSTM, and Transformer to demonstrate the generality of MIX. In total, we evaluate across 33 configurations, combining 11 datasets with 3 model architectures (see Sec. E.4 in Appendices for details).

**Baselines.** We compare our method with recent SOTA explanation approach for TSC including LIMESegment [76], InteDisUX [83], SpectralX [10] (see Sec. E.5 in Appendices for details).

**Evaluation Metric.** We evaluate explanations using four complementary criteria: First, we compute top-$k$ faithfulness by measuring the drop in confidence score when removing the $k$ most important features. These features are selected either using a greedy strategy or obtained from attribution maps. We report results for $k = 8$, 20, and 30 as **faithfulness@8, @20, and @30**. with MIX, since explanation output is for each view, we evaluate in all views then choose level with maximum average of those faithfulness. We report the performance of the single best-performing view for each dataset. Concretely, for each individual view, we average its performance across the different metrics (Faithfulness@8, @20, and @30). We then select the single view with the highest score and report its performance.This process highlights a key strength of our framework: its ability to identify the most effective explanatory perspective for a given task, rather than report the results by averaging. Second, we use **two faithfulness-oriented metrics** proposed in [84]: AUC$\tilde{S}_{top}$ and F1$\tilde{S}$. Third, we evaluate robustness by computing the **Jaccard coefficient** between two sets of top-30 features: one derived from the original input **x** and one from a noisy version of **x** (perturbed with small Gaussian noise). Fourth, for datasets with explanation ground truth, we compute **AUPRC and the Jaccard coefficient** between the top-$k$ features identified by the explanation and the annotated ground truth features. For synthetic and MIT-BIH datasets, we use our greedy selection scheme (Phase 3), which provides an even more direct comparison. In this phase, the most important features are collected from all views and then projected onto a single user-specified view (chosen as $cA_0$). The final explanation's quality is then evaluated on this chosen view only. Full details of all metrics are in Sec. E.6 in Appendices.

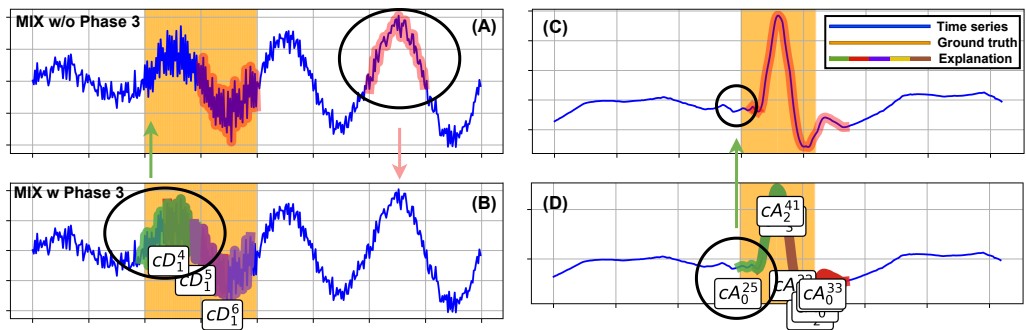

Figure 3: MIX explanations on the synthetic dataset (A, B) and MIT-BIH (C, D), without (A, C) and with (B, D) Phase 3. Labels use the format $cA_{\text{level}}^{\text{segment id}}$. Phase 3 enhances interpretability by highlighting key features of time series and relevant granularity.

## 3.2 Main results

**Results on synthetic and MIT-BIH dataset.** We report AUPRC and Jaccard Coefficient for our method compared to LIMESegment [76] and InteDisUX [83] on both the synthetic dataset and MIT-BIH. SpectralX [10] is not included to comparison as it cannot generate explanations in the raw time domain from spectrogram. As shown in Table 2, our framework outperforms both baselines. Additionally, the proposed Phase 3 boosts the Jaccard Coefficient from 0.6133 to 0.7481 on the synthetic dataset, and from 0.5946 to 0.6329 on MIT-BIH. The visualization examples in Fig. 3 show that multi-view greedy selection helps MIX identify more relevant features by exploring multiple wavelet levels, rather than relying solely on a single view.

**Results on UCR datasets.** The scatter plots in Fig. 2 compare our method with three state-of-the-art baselines including LIMESegment, InteDisUX, and SpectralX in terms of faithfulness and robustness. In the lower scatter plot, the vertical axis shows the average of Faithfulness@8, @20, and @30, while the upper plot displays robustness. Addi-

Table 2: Comparison on synthetic and MIT-BIH datasets.

| Method | Synthetic | | MIT-BIH | |
|---|---|---|---|---|
| | AUPRC | Jaccard | AUPRC | Jaccard |
| LIMESegment | 0.6663 | 0.1427 | 0.5651 | 0.3376 |
| InteDisUX | 0.6781 | 0.2077 | 0.6364 | 0.4261 |
| Ours | **0.7103** | 0.6133 | **0.7404** | 0.5946 |
| Ours w/ Greedy | - | **0.7481** | - | **0.6329** |

tionally, we present the average performance across 11 datasets using a grouped box plot in Fig. 2, which includes error bars for each metric. Across all evaluations, MIX consistently outperforms the baselines on both faithfulness and robustness. Our method achieves the best performance in **24/33** cases for faithfulness and **27/33** cases for robustness, outperforming the baselines across both criteria. We also compare $AUC\tilde{S}_{top}$ and $F1S$ compared to LIMESegment, IntedisUX with the results are in spider graph in Fig. 2 . Our method is still better than other two methods. More results are in Sec. F in Appendices.

## 3.3 Ablation Study

**Study on Cross-View Refinement.** To study the effect of cross-view refinement, we compare faithfulness scores between Phase 1 (without cross-view refinement) and Phase 2 across all views (represented by wavelet levels) in three different dataset/architecture setups: Yoga/BiLSTM, MSR-Train/Transformer, and Wafer/ResNet-34. The results show that cross-view refinement consistently improves faithfulness across all three setups indicates in Fig. 4 (A, B, C).

**Study on Our Attribution Mechanism.** To show the effectiveness of our attribution mechanism, we compare its performance with the standard IG method for each feature in view $V_r$ across three setups: Yoga/BiLSTM, Wafer/Transformer, and GPGender/ResNet-34. Our study shows that our attribution mechanism has higher faithfulness compared to the IG baseline indicates in Fig. 4 (D, E, F).

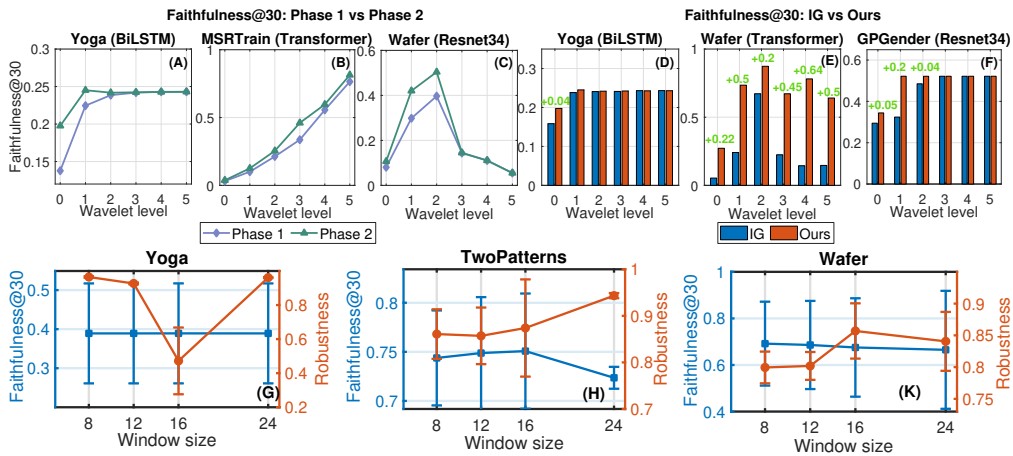

Figure 4: Ablation study on Cross-view Refinement (A, B, C), MIX's attribution mechanism (D, E, F), and window size, step size (G, H, K).

**Study on Window Size and Step Size.** We also study the effect of window size $w$ and step size $\delta = w/2$ on the performance of MIX using three datasets: *Yoga*, *TwoPatterns*, and *Wafer*. The results are presented in Fig. 4 (G, H, K). We observe that faithfulness remains relatively stable as $w$ increases, with some datasets exhibiting a peak at a specific value, such as $w = 16$ for the Wafer dataset. In addition, robustness is generally more stable across different window sizes but may peak and then slightly decline, as also observed on Wafer.

## 4 Related Works

Time series classification approaches can be broadly categorized into two types: **ante-hoc** and **post-hoc**. Ante-hoc approaches are interpretable by design, where the model's internal structure inherently provides explanations such as decision trees, shapelet-based models [92, 47, 22, 61, 24, 91, 18, 15, 28, 51, 88, 85, 54, 25, 26, 90], attention mechanisms [32, 31, 69, 9, 46], Symbolic Aggregate approXimation (SAX) [52, 71, 78, 41], prototypes [57, 21], features [87, 20, 35, 72, 93, 64, 6, 19, 30, 34, 60, 23], or other ways such as patches [56, 55]. In contrast, post-hoc approaches are applied after model training, typically when the model itself is a black box and not inherently interpretable. Post-hoc methods aim to explain the model's decision behavior after training. These can be further grouped into: (i) attribution-based methods, which assign importance scores to individual time points [80, 89, 68, 3, 95, 94, 50, 77, 73, 97, 70, 75, 62, 74, 58, 1, 65, 82, 43, 53, 5, 36, 86, 12, 17, 49, 48, 66, 11, 4, 16] or segments[29, 63, 76, 83]; (ii) subsequence-based methods, which identify discriminative time intervals [8, 33, 27]; and (iii) instance-based methods, which explain predictions by referencing similar or counterfactual instances [39, 14, 37, 2, 40], and other approaches such as using prototypes [13, 81, 96].

Our work is closely related to post-hoc segment-based attribution methods, which assign importance score to segments rather than time points [76, 83]. In the context of multi-view time-frequency explanations, MIX is most similar to SpectralX [10], which operates on a single time-frequency representation.

## 5 Conclusion

In this paper, we introduce a novel explainability framework for time series classification (TSC) based on the concept of multi-view. To the best of our knowledge, our work is the first to not only generate independent explanations across multiple views, but also to improve explanations across all views through a new cross-view refinement scheme. In addition, we propose a novel multi-view traversal greedy strategy that selects the most important features within the user's preferred view, further improve the explanation with constraints from human perspective.

## Acknowledgements

We special thank Hoang Anh Dau, Anthony Bagnall, Kaveh Kamgar, Chin-Chia Michael Yeh, Yan Zhu, Shaghayegh Gharghabi, Chotirat Ann Ratanamahatana, Eamonn Keogh and others for producing and maintaining the UCR and UEA time series archive, which we use in our work. We thank Owen Queen, Thomas Hartvigsen, Teddy Koker, Huan He, Theodoros Tsiligkaridis, Marinka Zitnik for providing the MIT-BIH dataset with explanation ground truth in their NeurIPS 2023 paper. We thank the anonymous reviewers for their valuable comments. This research is part-funded by the European Union (Horizon Europe 2021-2027 Framework Programme Grant Agreement number 10107245; views and opinions expressed are however those of the author(s) only and do not necessarily reflect those of the European Union. The European Union cannot be held responsible for them) and by the Engineering and Physical Sciences Research Council under grant number EP/X029174/1.

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

## Technical Appendices and Supplementary Material

In the appendices, we first provide mathematic notions with definition in Section A. In addition, we provide extensive background on time series classification (TSC), explanation methods for TSC, and the Discrete Wavelet Transform (DWT) in Sec. B.

We also include an in-depth analysis of the attribution mechanism used in SpectralX, focusing on its stability and the risk of false negative feature selection in Sec. C. This analysis motivates our choice to adapt Integrated Gradients (IG) as the foundation for attribution mechanism in MIX. Furthermore, we present insight analysis into our proposed variants, IGV and KIGV in Sec. D, to support their design and behavior. Furthermore, we include comprehensive experimental results and comparisons in Sec. F, along with full experimental setup details to support reproducibility in Sec. E.

To analyze insight of our method in various aspects, we conduct analysis in Section G. First of all, we analyze system design, then we move to extensive ablation studies to verify contribution of each core component. Finally, we analyze hyperparameters analysis and computational complexity of MIX compared to InteDisUX and SpectralX. We also include qualitative results in Sec. H.

We also put two discussion section, Section I and Section J to clearify the difference, possitive side and limitation when comparing post-hoc and ant-hoc explanation setting, local vs. global explanation, and multi-view with cross-modal aspects. To ensure transparency, we also dedicate two sections to discussing the limitations in Sec. K and broader impact of our method in Sec. L.

## Contents

## A   Table of Symbols

We define key notations in the paper in the Table 3.

## B   Background

### B.1   Problem Definition

**Definition B.1** (Deep Learning Model for TSC). A deep learning model for time series classification is a function $F_\theta : \mathbb{R}^{T \times d} \to [0, 1]^{|\mathcal{Y}|}$ parameterized by $\theta$, which maps an input time series $\mathbf{x} \in \mathbb{R}^{T \times d}$ to a probability distribution over a finite set of class labels $\mathcal{Y} = \{1, 2, \dots, C\}$, where $C = |\mathcal{Y}|$ is the number of classes. The output vector $f_\theta(\mathbf{x}) = [p_1, p_2, \dots, p_C]$ satisfies $\sum_{i=1}^{C} p_i = 1$ and $p_i \in [0, 1]$.

The model is trained by minimizing the cross-entropy loss over a labeled dataset $D = \{(\mathbf{x}_j, y_j)\}_{j=1}^{N}$, defined as:

$$\mathcal{L}_{CE}(\theta) = -\frac{1}{N} \sum_{j=1}^{N} \log F_\theta(\mathbf{x}_j)_{y_j},$$

where $f_\theta(\mathbf{x}_j)_{y_j}$ denotes the predicted probability for the true class $y_j$.

### B.2   Discrete Wavelet Transform

We begin by introducing the Continuous Wavelet Transform (CWT), Discrete Wavelet Transform (DWT), and the Haar wavelet:

**Definition B.2** (Continuous Wavelet Transform [7]). The *Continuous Wavelet Transform (CWT)* of a time series signal $\mathbf{x}$ is defined as:

$$H(\mathbf{x}, \zeta, \tau) = \frac{1}{\sqrt{|\zeta|}} \int x(t) \cdot \psi^* \left( \frac{t - \tau}{\zeta} \right) dt$$

Table 3: Notation used throughout the paper.

| Symbols | Meaning |
|---|---|
| $\mathbf{x} = \{x_1, x_2, \ldots, x_T\} \in \mathbb{R}^{T \times d}$ | A time series vector of dimension $d$ and $T$ time steps. |
| $D = \{(\mathbf{x}_i, y_i)\}_{i=1}^{N}$ | TSC dataset |
| $F : \mathbb{R}^T \to \mathbb{N}$ | Deep learning model |
| $V : \mathbb{R}^d \to \mathbb{R}^{d'}$ | View |
| $V_r$ | $r$th view |
| $V^{-1} : \mathbb{R}^{d'} \to \mathbb{R}^d$ | reverse view |
| $F_V = F \circ V^{-1}$ | New model combine DL model with view for input as view |
| $se^i = se_{t_{\text{start}}}^{t_{\text{end}}}$ | $i$th Segment with start time step is $t_{\text{start}}$ and end time step is $t_{\text{end}}$ |
| $S = \{se^1, se^2, \ldots, se^M\}$ | Set of segments |
| $s_{se^i}$ | score of segment $se^i$ |
| $\mathcal{E}(\mathbf{x}, F, S) = \{s_{se^i} | se^i \in S\}$ | Set of score for each segment |
| $S_V$ | Set of views |
| $\mathcal{E}_V(\mathbf{x}, F, S, V) = \{s_{se^i} \mid se^i \in S\}$ | Set of score for each segment for the corresponding view |
| $cA_r$ | $cA$ on $r$th level |
| $cD_r$ | $cD$ on $r$th level |
| $OS_{cA_r}$ | Set of overlapping segments on $cA_r$ |
| $OS_{cD_r}$ | Set of overlapping segments on $cA_r$ |
| $S_r = OS_{cA_r} \cup OS_{cD_r} \cup \ldots \cup OS_{cD_1}$ | Set of overlapping segments of $r$th view |
| $\text{IG}_j(\mathbf{x})$ | Integrated Gradients of $\mathbf{x}$ |
| $\text{IGV}_j(V(\mathbf{x}), V(\mathbf{x}'))$ | Integrated Gradients of a view of $\mathbf{x}$ |
| $\text{OSIGV}(se^i)$ | Integrated Gradients of segments of a view of $\mathbf{x}$ |
| $\text{KIG}_i(V(\mathbf{x}), V(\mathbf{x}'))$ | Keystone-first Integrated Gradients of a view of $\mathbf{x}$ |
| $AUC\tilde{S}_{\text{top}}$ | Metric in [84] |
| $KAUC\tilde{S}_{top}$ | Keystone $AUC\tilde{S}_{\text{top}}$ |

where $H(\mathbf{x}, \zeta, \tau)$ denotes the wavelet coefficient as a function of the time series $\mathbf{x}$, the time-shift parameter $\tau$, and the scale parameter $\zeta$; $\psi$ is the *mother wavelet* (or basis function), and $\psi^*$ is its complex conjugate. The function $x(t)$ is the input signal evaluated over continuous time, we can denote $x(t) = x_t$.

The oldest and simplest wavelet is the *Haar wavelet*, whose mother wavelet is defined as:

$$\psi(t) = \begin{cases} 1 & \text{if } 0 \leq t < \frac{1}{2} \\ -1 & \text{if } \frac{1}{2} \leq t < 1 \\ 0 & \text{otherwise} \end{cases}$$

The Continuous Wavelet Transform (CWT) is computationally expensive. The Discrete Wavelet Transform (DWT) offers a more efficient alternative by applying high-pass and low-pass filters across multiple scales [7]. DWT has proven effective in time series analysis and naturally produces hierarchical, interpretable multi-resolution representations. However, its potential for explainability in TSC XAI remains underexplored.

## C SpectralX Attribution Mechanism Analysis

Table 4: Faithfulness@30 and robustness scores for two random seeds across datasets.

| Dataset | Seed 1 Faithfulness@30 | Seed 2 Faithfulness@30 | Jaccard |
|---|---|---|---|
| CinCECGTorso | $0.3850 \pm 0.2986$ | $0.3709 \pm 0.2828$ | $0.4678 \pm 0.2671$ |
| MixedShapesRegularTrain | $0.5036 \pm 0.1730$ | $0.5283 \pm 0.2371$ | $0.4834 \pm 0.1557$ |

### C.1 Stability

According to the SpectralX method described in Section E.5, we hypothesize that its random feature selection process may be sensitive to the randomness of the system. To investigate this, we run

SpectralX with the same algorithmic setup but vary only the manual seed. We then compare the resulting explanations using faithfulness@30 and the Jaccard coefficient between the top-30 selected features under each setup. The results, shown in Table 4, indicate that while the overall faithfulness remains comparable, the Jaccard coefficient is only around 0.4. This suggests that the set of top important features can change significantly due to randomness. While this does not negate its overall faithfulness, it makes the explanation unstable.

### C.2 False Negative Analysis

According to the way spectralX generate masks in Sec. E.5, the all $P$ masks may never unmask a specific features. Formally, given a feature $f_i$, then probability that all $P$ masks never unmask $f_i$ is larger than zero. The probability is calculate as follow:

**Theorem C.1.** *Given the spectralX random maske generation process with $P$ masks, $R$ features unmasked each mask, and time series length $T$, then the probability that a feature $f_i$ have never been unmasked in all $P$ masks is:*

$$Pr = (1 - R/T)^P$$

*Proof.* Overall, with $T$ features, when generating maskes, we have total combination of $T$ choose $R$: $\binom{T}{R}$

Assume that the feature $f_i$ has never been unmasked, then each time genration process only choose $R$ features in total $T - 1$ features. Then, total number of that is: $\binom{T-1}{R}$.

Then, probability for a mask does not unmask $f_i$ is:

$$
\begin{aligned}
pr &= \frac{\binom{T-1}{R}}{\binom{T}{R}} \\
&= \frac{\frac{(T-1)!}{R!(T-1-R)!}}{\frac{T!}{R!(T-R)!}} \\
&= \frac{(T-1)! \cdot R!(T-R)!}{T! \cdot R!(T-1-R)!} \\
&= \frac{(T-R)}{T} \\
&= 1 - \frac{R}{T}
\end{aligned}
$$

For $P$ masks, then the probability that all $P$ masks never unmask $f_i$ is:

$$Pr = pr^P = \left(1 - \frac{R}{T}\right)^P$$

$\square$

The case when a feature $f_i$ is never unmask in $P$ masks can result that importance score for this is zero just by sampling but not actual influence to the model's decision. With the default setup of SpectralX, where $R = 10$ and $P = 2000$, the probability $Pr$ is approximately $4 \times 10^{-5}$ when $T = 2000$. However, this value increases with larger $T$; for example, when $T = 10000$, we have $Pr \approx 0.135$. In practice, SpectralX remains effective because this probability is still relatively small. Nonetheless, the growth of $Pr$ with respect to $T$ highlights a limitation of SpectralX when applied to longer time series.

## D   Analysis on Attribution Mechanism of MIX

**IGV.** Our IGV is just modified version of IG for new function $F \circ V^{-1}$ on $V(\mathbf{x})$, then it still staisfies axioms of IG in [80].

## D.1 Invariant to Implementation.

Our attribution mechanisms, IGV and KIGV, are fully deterministic and do not rely on any randomized operations. Given a window size $w$ and step size $\delta$, the overlapping segments generated in all views are fixed for each dataset. Consequently, both OSIG and OSIGV produce deterministic results. In the $\text{KAUC}\tilde{S}_{top}$ criterion, we use a baseline $x'$ defined as the mean of each view, which introduces no randomness. As a result, the adaptive selector also operates independently of noise. Therefore, given a model $F$ and an input time series $x$, MIX produces explanations that are invariant to implementation and do not depend on any random process. This ensures that our framework remains stable and consistent, even in the presence of system-level randomness.

## D.2 Analysis on Keystone-first Integrated Gradients

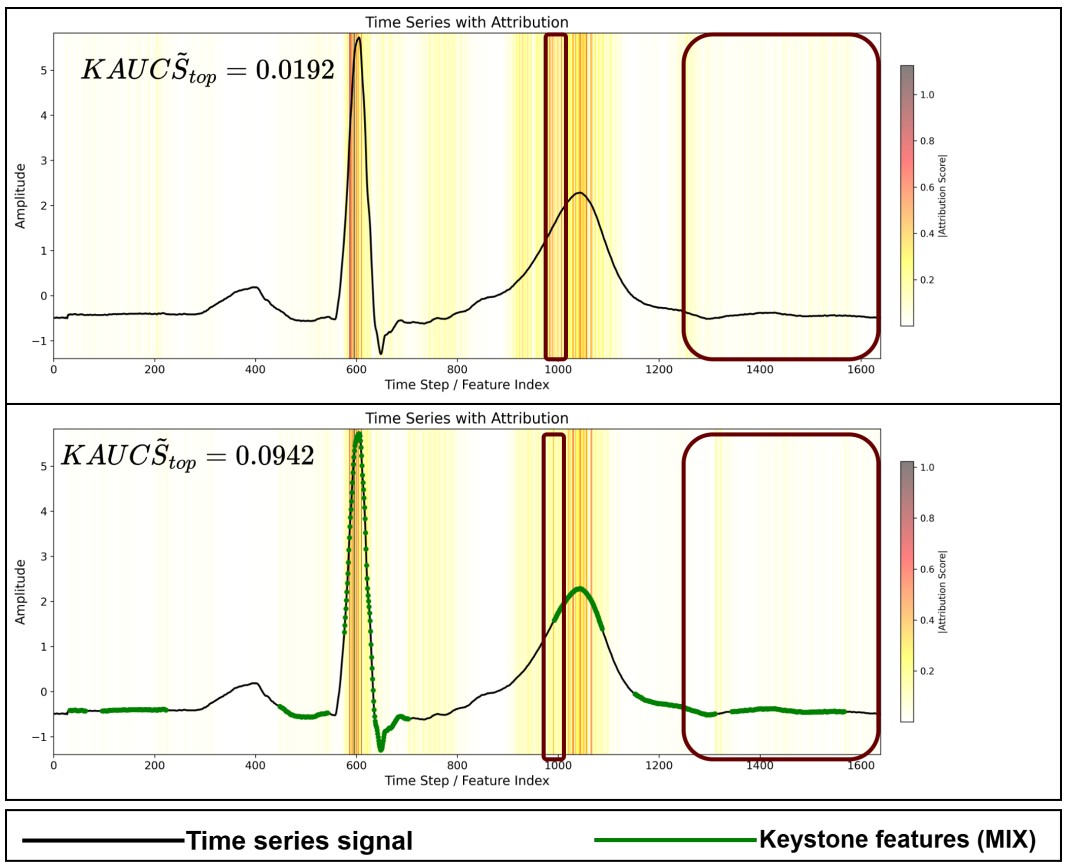

Figure 5: Comparison of KIGV (bottom) with conventional Integrated Gradients (IG, top) on the CinCECGTorso dataset. The time series signal is shown as a black line, while the highlighted segment in green indicates the keystone parts used as input for KIGV. The background color represents the attribution value at each time step. In the regions highlighted by red rectangles, KIGV avoids attributing importance to less relevant features that are mistakenly emphasized by standard IG. This demonstrates the effectiveness of cross-view refinement in filtering out irrelevant features. It is also reflected by higher $KAUC\tilde{S}_{top}$ with KIGV.

In both the main paper and Appendices, we have shown that IGV, satisfies the axioms of Integrated Gradients (IG) and our overall attribution mechanism is implementation invariant. Here, we further analyze KIGV within the cross-view refinement framework to demonstrate its advantages over standard IG. By prioritizing the explanation of "keystone" features, KIGV effectively reduces the noise present in importance scores generated by IG, resulting in more accurate attribution maps. As a result, OSIGV also benefits from these improved scores. As illustrated in Fig. 5, our method

avoids highlighting irrelevant features, which is further supported by the higher $KAU\tilde{CS}_{top}$ values achieved with KIGV compared to IG.

# E Experimental Setup

## E.1 More Details about Datasets

**ArrowHead.** The ArrowHead dataset comprises univariate time series derived from the outlines of arrowhead images, converted into time series using an angle-based method. It contains 36 training instances and 175 testing instances, each of length 251, categorized into three classes: "Avonlea," "Clovis," and "Mix." This dataset is particularly relevant in anthropology for classifying projectile points based on shape distinctions, such as the presence and location of notches.

**Strawberry.** The Strawberry dataset consists of univariate time series data representing different classes of strawberries. It includes 613 training instances and 370 testing instances, each of length 235, divided into two classes. This dataset is useful for classification tasks involving agricultural products, such as distinguishing between different strawberry varieties or assessing quality.

**Yoga.** The Yoga dataset features univariate time series data related to yoga poses, aiming to classify different postures or movements. It comprises 300 training instances and 3,000 testing instances, each of length 426, categorized into two classes. This dataset is valuable for applications in human activity recognition and pose classification.

**FordA.** The FordA dataset contains univariate time series data collected from automotive subsystems, with the task of classifying whether a symptom exists under typical operating conditions. It includes 1,320 training instances and 1,320 testing instances, each of length 500, divided into two classes. This dataset is pertinent for fault detection and predictive maintenance in automotive systems.

**FordB.** Similar to FordA, the FordB dataset comprises univariate time series data from automotive subsystems. However, the test data samples in FordB were collected under noisy conditions, adding complexity to the classification task. It consists of 3,636 training instances and 810 testing instances, each of length 500, categorized into two classes. This dataset challenges models to maintain accuracy despite increased noise in the data.

**MixedShapesRegularTrain.** The MixedShapesRegularTrain dataset consists of univariate time series obtained by converting two-dimensional shapes into one-dimensional time series. It includes 500 training instances and 2,425 testing instances, each of length 1,024, divided into five classes: Arrowhead, Butterfly, Fish, Seashell, and Shield. This dataset is designed to test the ability of classifiers to distinguish between different shape patterns.

**CinCECGTorso.** The CinCECGTorso dataset is a multivariate time series dataset containing electrocardiogram (ECG) data collected from the torso. It comprises 40 training instances and 1,380 testing instances, each of length 1,639, categorized into four classes. This dataset is used for classifying different heart conditions and is relevant in the field of biomedical signal processing.

**GunPointMaleVersusFemale.** The GunPointMaleVersusFemale dataset is a remake of the original GunPoint dataset, featuring recordings from a male and a female actor performing gun and point gestures. It includes 135 training instances and 316 testing instances, each of length 150, divided into two classes. This dataset is useful for studying variations in human motion patterns across different individuals.

**TwoPatterns.** The TwoPatterns dataset is a synthetic univariate time series dataset designed to test the ability of classifiers to distinguish between different types of patterns. It consists of 1,000 training instances and 4,000 testing instances, each of length 128, categorized into four classes. This dataset is commonly used for benchmarking time series classification algorithms.

**MixedShapesSmallTrain.** Similar to MixedShapesRegularTrain, the MixedShapesSmallTrain dataset comprises univariate time series obtained by converting two-dimensional shapes into one-dimensional time series. However, it has a smaller training set with 100 training instances and 2,425 testing instances, each of length 1,024, divided into the same five classes: Arrowhead, Butterfly, Fish, Seashell, and Shield. This dataset tests the performance of classifiers with limited training data.

**Wafer.** The Wafer dataset contains univariate time series data from semiconductor manufacturing processes, used to detect whether a wafer is normal or exhibits a fault. It includes 1,000 training

instances and 6,164 testing instances, each of length 152, categorized into two classes. This dataset is pertinent for quality control and fault detection in industrial settings.

**MIT-BIH.** The MIT-BIH dataset has ECG recorded from 47 subjects at the sampling rate 360Hz. The raw dataset was then window-sliced into 92511 samples of 360 timestamps each. Two cardiologists have labeled each beat independently. The dataset is choose left bundle branch block beat (L), and right bundle branch block beat (R) as anomaly with label 1 and normal as label 0.

## E.2  Data Normalization and Split Mechanism

We apply Z-score normalization to preprocess the data before training and testing deep learning models. The dataset is divided into three parts: 80% for training, 10% for validation, and 10% for testing. We conduct explanation on testing data in all setups.

## E.3  Training details for Classifiers

For UCR time series datasets, we use a batch size of 16, cross-entropy loss, and the Adam optimizer with a learning rate of $2 \times 10^{-4}$ for 200 epochs. For the synthetic dataset, we use a batch size of 64, cross-entropy loss, and the Adam optimizer with a learning rate of 0.001 for 10 epochs, using a simple 1D CNN architecture. For MIT-BIH, we adopt the CNN architecture from [66, 48], use PolyLoss [42], a batch size of 16, the Adam optimizer with a learning rate of $2 \times 10^{-4}$, and train for 200 epochs.

## E.4  DL Architecture

We use the same architecture with SpectralX for TSC on 11 UCR datasets. We use simple 1D CNN for synthetic dataset and MIT-BIH.

## E.5  Baselines

**LIMESegment.** LIMESegment is a post-hoc TSC method based on the LIME explanation method on distinct segments, which is generated from time series using the NNSegment method. LIMESegment uses NNSegment using input as a time series $\mathbf{x}$, a predefined number of change points $T'$, window size $ws$, then generates segments based on the time series itself. After that, LIMESegment uses LIME to explain segments with input as a time series, a set of segments, and the model.

**InteDisUX.** InteDisUX is a new post-hoc segments-based TSC explanation approach using model's information to generate set of segments for explanation. Firstly, InteDisUX initialize $N_S$ equal-length segments. Then, the algorithm iteratively merge every two consecutive segments pair using faithfulness gain and discrimination gain metrics.

**SpectralX.** The method explains time series in the time-frequency domain by transforming the time series to a spectrogram using STFT, then applying a greedy selection on that, with each feature defined as a time-frequency region. This greedy strategy iteratively estimates deletion and insertion scores and chooses a feature with the best score until it reaches the desired number of features. During the insertion/deletion phase, SpectralX generates diverse perturbations by selecting random features. Technically, given predefined $R$ time-frequency regions unmasked, spectralX generates $P$ masks that have $R$ regions unmasked, then the insertion/deletion score of a feature is estimated as the reduction of confidence score when inserting/deleting that feature.

## E.6  Evaluation

**Faithfulness.** Following the SpectralX paper [10], we evaluate the faithfulness of explanations by removing the top-$k$ features and measuring the reduction in the model's confidence for the ground truth label. We use $k = 8, 20, 30$, which represents a sufficient number of features to significantly influence the model's decision. Then, the faithfulness metrics is: faithfulness@8, faithfulness@20, faithfulness@30. For each individual view, we average its performance across the different metrics (Faithfulness@8, @20, and @30). We then select the single view with the highest score and report its performance. This process highlights a key strength of our framework: its ability to identify the most effective explanatory perspective for a given task, rather than report the results by averaging.

To compare with SpectralX, we need to convert segment-level explanation scores into individual feature-level scores as presented in Section 2.4. The score for each time point $x_j$ is calculated as the maximum absolute score of any segment that contains it. Then we follow the evaluation scheme of SpectralX: measuring Faithfulness@k by removing the top-k most important features and calculating the degradation in the model's confidence score. To compare with segment-based methods, we use the $AUC\tilde{S}_{\text{top}}$ and $F1\tilde{S}$ metrics proposed in [84]. They evaluate performance by measuring the area under the confidence degradation curve as an increasing fraction of the most important features (the top k-quantile) are removed. Furthermore, for synthetic and MIT-BIH datasets, we use our greedy selection scheme (Phase 3), which provides an even more direct comparison. In this phase, the most important features are collected from all views and then projected onto a single user-specified view (chosen as $cA_0$). The final explanation's quality is then evaluated on this chosen view only. Therefore, these ensure fair comparisons to single-view methods.

**Robustness.** Following the SpectralX paper [10], we evaluate robustness by measuring the overlap between the top-$k$ features identified with and without the addition of noise. Specifically, we use the Jaccard coefficient between these two sets as a quantitative measure of robustness. MIX follows standard practices for evaluating the stability of an explanation by measuring its consistency when the input is slightly perturbed. First, we take the original input time series x and generate an explanation, from which we identify the set of its top-k most important features E. Next, we create a perturbed version of the input x'. By adding a small amount of Gaussian noise. We then generate a new explanation for this noisy input, and identify its corresponding set of top-30 features, E'. Finally, we compute the Jaccard coefficient between these two sets of top-30 features: E, E'. Higher Jaccard score (greater overlap between the two sets) signifies that the explanation is stable and consistent. A lower score suggests the explanation is sensitive to minor perturbations and is therefore less robust.

**Relevance identification.** We follow paper [84] to evaluate our explanation by using their two proposed metrics called $AUC\tilde{S}_{top}$ and $F1S$. First, we define a *top-k-quantile-masked sample*, denoted $\tilde{\mathbf{x}}_k^{\text{top}}$, as a perturbed version of the input $\mathbf{x}$, where the top-$k$ most important features (based on absolute attribution scores) are replaced. Features are selected using a $(1-k)$-quantile threshold. [84] uses random noise for replacement.

The top-$k$-quantile strategy is typically applied to point-wise explanations. However, since segment-based attribution produces scores at the segment level, we assign importance scores to individual features in a view $V_r(\mathbf{x})$ by taking the maximum absolute score across all segments that include the feature:

$$s_j = \max \left\{ |s_{se^i}| \ \middle| \ V_r(x)_j \in se^i \right\}.$$

To quantify explanation quality, we define the normalized degradation score as the drop in model confidence when $\mathbf{x}$ is perturbed into $\tilde{\mathbf{x}}$:

$$\tilde{S}(\mathbf{x}) = \frac{S(\mathbf{x}) - S(\tilde{\mathbf{x}})}{S(\mathbf{x})},$$

where $S(\mathbf{x})$ is the model's confidence for the predicted class.

By progressively corrupting the top- or bottom-$k$ features, we construct $\tilde{S}$–$\tilde{T}$ curves, where $\tilde{T} = \frac{\bar{T}}{T}$ represents the fraction of corrupted time steps. The area under this curve quantifies overall degradation:

$$AUC\tilde{S} = \int_0^1 \tilde{S} \, d\tilde{T}.$$

In particular, $AUC\tilde{S}_{\text{top}}$ captures the degradation from top-$k$-quantile-masked samples, with $k \in [0, 1]$. A higher $AUC\tilde{S}_{\text{top}}$ indicates a more faithful explanation.

In terms of $F1S$, we compute $AUC\tilde{S}_{\text{bottom}}$, which measures the degradation when perturbing features with the *lowest* attribution scores. Specifically, features are perturbed if their scores fall below the threshold defined by the $k$-quantile, i.e., the bottom-$k$-quantile of importance scores. Then $F1S$ is calculated by:

$$F1S = \frac{AUC\tilde{S}_{top} \cdot (1 - AUC\tilde{S}_{bottom})}{AUC\tilde{S}_{top} + (1 - AUC\tilde{S}_{bottom})}$$

Since no evaluation metric is perfect, using two distinct, standard faithfulness metrics provides a more robust and trustworthy validation for MIX.

**Reasoning of Faithfulness Metrics.** Our method helps users understand which features are influential in a model's decision but does not prove a causal link, which is outside the scope of this study. Without a perfect causal method, the faithfulness metric is the community's best available proxy for evaluating post-hoc explanations. It directly and pragmatically tests the model's internal behavior: if the model's confidence drops significantly when we remove the features the explainer identifies as important, it provides strong evidence that the explanation has faithfully captured what the model considers important for its decisions. While it cannot reveal the underlying "logic" or "reason" why the model learned to depend on those features, it rigorously validates that our explanation is true to the black-box model's function. This is the standard and accepted goal of attribution-based methods in the current XAI landscape.

**Validity of the experiments.** The validity of our experiments is grounded in our use of standardized, community-accepted post-hoc evaluation protocols for TSC XAI [83, 10, 84]. *For dataset selection*, to ensure a direct and relevant comparison, we use 9 UCR datasets from SpectralX and 2 additional UCR datasets for comprehensive evaluations. *For evaluation metrics*, we adopted the evaluation protocol from SpectralX to ensure fair comparisons. For faithfulness, we follow their method of measuring the drop in model confidence after removing the top-k most important features. For robustness, we also follow their approach of measuring the overlap between the top features of an original instance and a noisy version using Jaccard Coefficient, which is similar to the way SpectralX quantifies the overlap between two fixed-size sets. We also apply $\text{AUC}\tilde{S}_{\text{top}}$ and $\text{F1}\tilde{S}$ from [84]. Since MIX generates local explanations for each sample, we calculate the faithfulness and robustness scores for every sample in the test set and report averaged values. This is the *standard practice* in post-hoc TSC XAI. We also provide a global explanation in Table 14.

### E.7 MIX Hyperparameters

For the top $h$, we use $h$ as $1/5$ of all segments for all dataset. For thresold for $KAU\tilde{S}_{top}$, we use $\kappa = 0.3$ for all setups. According to window size and step size, we adapt for each dataset as follow:

Table 5: Window size and step size settings for each dataset.

| Dataset | Window Size | Step Size |
|---|---|---|
| ArrowHead | 10 | 5 |
| Strawberry | 8 | 4 |
| Yoga | 8 | 4 |
| FordA | 10 | 5 |
| FordB | 12 | 6 |
| MixedShapesRegularTrain | 8 | 4 |
| CinCECGTorso | 24 | 12 |
| GunPointMaleVersusFemale | 8 | 4 |
| TwoPatterns | 8 | 4 |
| MixedShapesSmallTrain | 24 | 12 |
| Wafer | 12 | 6 |

### E.8 Hardware Details

We use the system with 1 GPU GeForce RTX 4070 in Ubuntu 20.04.6 LTS.

## F Extensive Experimental Results

### F.1 Results on UEA multivariate time series.

The Tab. 6 shows averaged results on UEA Basic Motion on 4 DL models (CNN, PatchTST, Transformer, ResNet34). MIX still outperforms IntedisUX as well as SpectralX.

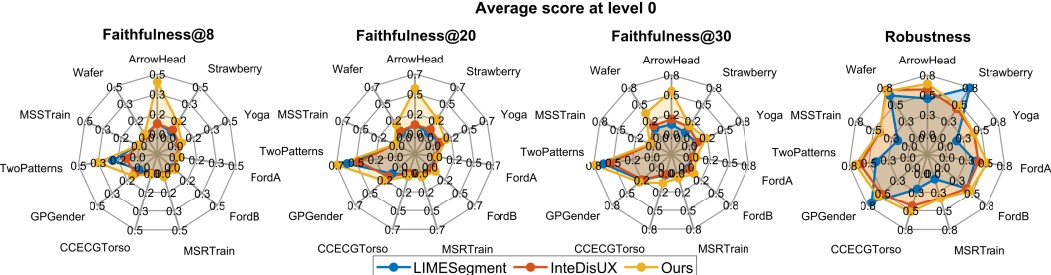

Figure 6: Faithfulness and Robustness of explanation of our method, LIMESegment and InteDisUX at view $cA_0$.

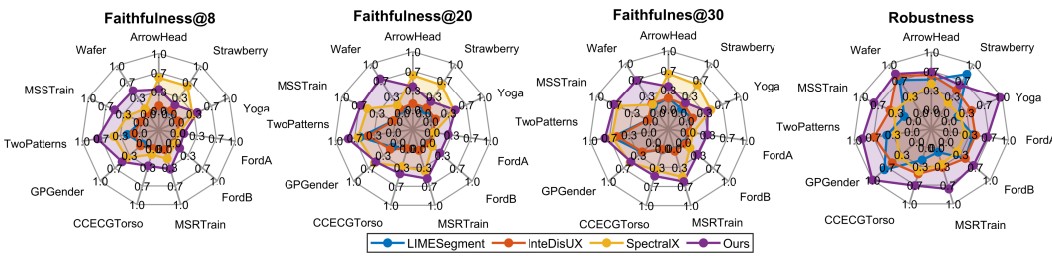

Figure 7: Comparison of Our method with LIMESegment, InteDisUX, and SpectralX, showing average performance of 3 DL architectures across 11 datasets .

## F.2 Extensive Comparison

**Compare in $cA_0$.** We compare our method with LIMESegment and InteDisUX in the same view, .i.e., $cA_0$, in Fig. 6, in terms of faithfulness@8, faithfulness@20, faithfulness@30 and robustness. The results show that MIX has significant higher faithfulness and robustness compared to LIMESegment and InteDisUX.

**Comparison average over architecture for each dataset.** We compare MIX with three baselines, which are LIMESegment, InteDisUX, and SpectralX, on faithfulness@8, @20, @30, and robustness across all UCR datasets. Results in Fig. 7 show that MIX consistently outperforms all baselines.

**Comparison on faithfulness of MIT-BIH dataset.** The Table 7 shows that MIX outperforms SpectralX for faithfulness@30 for MIT-BITH data.

**Average MIX vs. SpectralX comparison** To provide a more granular view of the results in Figure 2, the Figure 7 details the mean for each dataset for Faithfulness@8, Faithfulness@20, Faithfulness@30 and Robustness averaged over the three deep learning architectures for every dataset. We also provide mean and standard deviation for Faithfulness@30 comparing MIX vs. SpectralX in the Table 8.

**Comparison MIX vs SpectralX.** To compare with SpectralX, which also explains in the time-frequency domain, we present results in Fig. 8. In addition, Figure 8 show comparison of MIX vs SpectralX. Across the 11 UCR datasets, MIX outperforms SpectralX in terms of Faithfulness@8 on 9 datasets, Faithfulness@20 on 8 datasets, and Faithfulness@30 on 7 datasets, and Robustness on all datasets.

## G Extensive Analysis

### G.1 System Design Analysis

Our framework is built systematically from well-established principles, where each new component serves a specific purpose to address limitations in prior work. Please find below step-by-step justifications of our designs (IGV, OSIGV, KIGV, cross-view refinement).

**IGV.** Our method is built upon Integrated Gradients (IG) , a widely used and axiomatically sound attribution method. IGV, is the necessary mathematical adaptation of IG to operate in a transformed

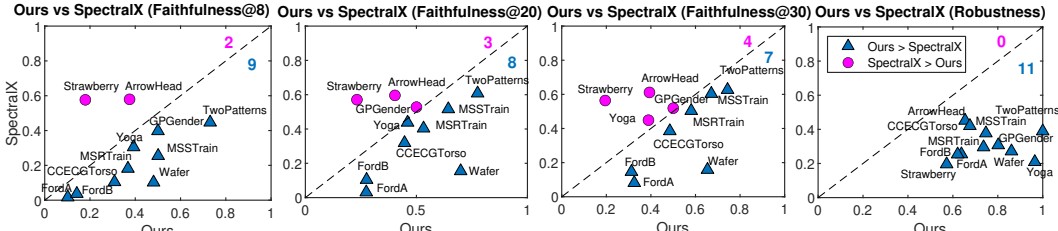

Figure 8: Comparison Ours vs SpectralX

Table 6: Performance Comparison on Multivariate UEA BasicMotions Dataset

| Metric | InteDisUX | SpectralX | MIX |
|--------|-----------|-----------|-----|
| Faithfulness@8 | 0.009 +/- 0.014 | 0.029 +/- 0.029 | **0.060 +/- 0.048** |
| Faithfulness@20 | 0.011 +/- 0.013 | 0.044 +/- 0.026 | **0.106 +/- 0.088** |
| Faithfulness@30 | 0.012 +/- 0.013 | 0.074 +/- 0.022 | **0.157 +/- 0.100** |
| AUCStop | 0.475 +/- 0.094 | - | **0.557 +/- 0.093** |
| F1S | 0.208 +/- 0.056 | - | **0.299 +/- 0.059** |

"view" space (e.g., the wavelet domain). As defined in our paper, if a view transformation $V$ is differentiable and invertible, IGV is simply IG applied to the composite model $F \circ V^{-1}$. This provides a mathematically sound way to calculate feature importance in any given view, not just the input space.

**OSIGV.** To make the point-wise scores from IGV more aligned with human perception, we aggregate them into segments. Our method OSIGV (Overlapping Segment-level Integrated Gradients for a View) is designed to improve upon prior segment-based approaches and make explanations more robust and interpretable. First, we use IGV to calculate an importance score for each individual point in a given view V(x). Second, we generate a set of overlapping segments across the view, where each segment is defined by a window size. The final score for each segment is then calculated as the sum of the IGV scores of all the individual points contained within that segment, as defined in our paper. The use of overlapping segments is a key design choice that directly addresses a limitation in prior work like InteDisUX, which uses non-overlapping segments. Non-overlapping segments risk splitting meaningful temporal patterns and can miss important information that occurs at the boundaries between segments.

**Interactive Refinement (KIGV and Cross-View Refinement).** The perceived complexity of our framework arises from our main contribution: cross-view refinement, which is enabled by our proposed Keystone-first Integrated Gradients (KIGV) method. Our motivation comes from the "keystone species" concept in ecology; just as a keystone species has a disproportionate impact on its environment, we hypothesize that certain "keystone features" in one view are strongly indicative of the model's core reasoning . Our novel KIGV method leverages this idea by first attributing importance to these keystone features, which are approximated as the top-h segments from the best view identified in Phase 1. By focusing the IG calculation path through these important regions first, we can generate a more robust gradient signal with less noise, and this refined attribution map is then used to improve the explanations in other views. To ensure that the cross-view refinement step is always beneficial, we introduce a final safeguard: the adaptive selector. The refined explanation generated by KIGV is only adopted for a given view if it shows a demonstrable improvement over the original one from Phase 1. This ensures our "heuristic" choice is not arbitrary but is instead a principled decision controlled by an objective quality metric based on paper [84].

**Scores for segments.** While time point-based explanations are common, they can be fragmented and difficult for users to interpret. Providing scores for contiguous segments is often more meaningful because it is better aligned with human perception as pointed out in [80]. For instance, on MIT-BIH ECG, a cardiologist can see that an entire heartbeat segment is responsible for an anomalous prediction. This is significantly more actionable and interpretable than analyzing hundreds of individual, high-importance time points, helping experts to easily validate whether a model's reasoning aligns with their own medical knowledge.

Table 7: Comparison of Faithfulness@30 metric scores between MIX and SpectralX on MIT-BIH dataset.

| Metric | IntedisUX | SpectralX | MIX |
|---|---|---|---|
| Faithfulness@30 | 0.6656 | 0.7478 | **0.8371** |

Table 8: A comparison of MIX and SpectralX performance across various datasets. The values represent mean $\pm$ standard deviation. Bold indicates the superior result for each dataset.

| Dataset | MIX | SpectralX |
|---|---|---|
| ArrowHead | $0.394 \pm 0.357$ | $\mathbf{0.6112 \pm 0.3778}$ |
| Strawberry | $0.1951 \pm 0.1913$ | $\mathbf{0.5638 \pm 0.3732}$ |
| Yoga | $0.3891 \pm 0.1280$ | $\mathbf{0.4486 \pm 0.0266}$ |
| FordA | $\mathbf{0.3254 \pm 0.1382}$ | $0.0813 \pm 0.0163$ |
| FordB | $\mathbf{0.3128 \pm 0.1401}$ | $0.1463 \pm 0.0633$ |
| MixedShapesRegularTrain | $\mathbf{0.5808 \pm 0.2114}$ | $0.5036 \pm 0.1730$ |
| CinCECGTorso | $\mathbf{0.4849 \pm 0.2829}$ | $0.3850 \pm 0.2986$ |
| GPGender | $0.5003 \pm 0.0345$ | $\mathbf{0.5190 \pm 0.0607}$ |
| TwoPatterns | $\mathbf{0.7438 \pm 0.0484}$ | $0.6266 \pm 0.0748$ |
| MixedShapesSmallTrain | $\mathbf{0.6710 \pm 0.0826}$ | $0.6036 \pm 0.0826$ |
| Wafer | $\mathbf{0.6538 \pm 0.1417}$ | $0.1585 \pm 0.0954$ |

**Circular Dependency Justification.** First, in Phase 1, the framework generates an initial, independent explanation for every view using the OSIGV attribution mechanism. This step is foundational, creating a complete set of unrefined explanations. At this stage, each explanation is generated in isolation based solely on its own view, and no refinement or "best view" selection has yet occurred. This provides the baseline set of explanations upon which the next phase will operate. MIX then moves to Phase 2, which is a two-step procedure that clearly separates selection from refinement. In the first step, the quality of the initial explanations from Phase 1 is evaluated using the $KAUC\tilde{S}_{top}$ score. The view that achieves the highest score is designated as the "best view" ($V$). Only *after* this selection is complete, does the refinement process begin. The information from the now-selected $V$ is used as a static guide to refine the explanations of the other views using the Keystone-first IG (KIGV) method. In summary, the selection of the best view depends on the quality of the initial, unrefined explanations, and the refinement process depends on the result of that selection. The refined explanations do not influence the initial scores that were used to make the selection in the first place. This strictly linear process ensures that there is no circular logic.

In summary, MIX's design is not complex for its own sake, but is a direct result of our attempt to solve several key challenges of this novel multi-view XAI setup at once. To create a robust multi-view explanation system, our framework was built to simultaneously address: providing an axiomatically-grounded explanation for any given view space (handled in Phase 1), Enabling views to interact and demonstrably improve one another's faithfulness (the core of Phase 2), Ensuring the final, user-facing explanation is of the highest possible quality (the goal of the Adaptive Selector in Phase 2 and the Greedy Selection in Phase 3).

### G.2 Extensive Ablation Studies

Along with ablation studies on overall Phase 2, the attribution mechanism (compared to standard IG), and the effects of window and step size, we also investigate the adaptive selector, which is a novel component in cross-view refinement. The adaptive selector dynamically determines whether to adopt the newly generated explanation or retain the previous one, thereby maintaining the quality of the explanations. Our ablation results, shown in Fig. 9, demonstrate that the adaptive selector consistently enhances explanation quality across all wavelet levels on both evaluated datasets.

### G.3 Hyperparameters Sensitivity

We will provide analysis how we choose hyperparameters and sensitivity analysis on that.

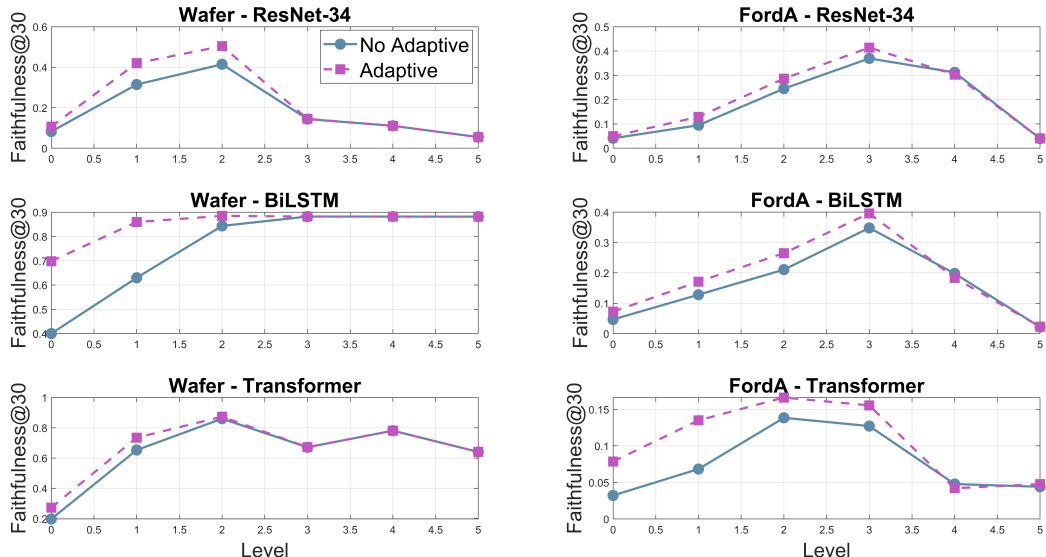

Figure 9: Ablation study on the adaptive selector for the Wafer and FordA datasets across three deep learning architectures. The results show that incorporating the adaptive selector (adaptive line) improves explanation quality across all wavelet levels compared to using KIGV alone (no adaptive line).

Table 9: Hyperparameter sensitivity for window and step size on the FordA dataset. Faithfulness is measured at k=30 (Faithfulness@30).

| Window Size | Step Size | Faithfulness@30 | Robustness |
|:-----------:|:---------:|:---------------:|:----------:|
| 8 | 4 | $0.3238 \pm 0.1445$ | $0.6527 \pm 0.1054$ |
| 16 | 8 | $0.2480 \pm 0.0661$ | $0.6647 \pm 0.1350$ |
| 32 | 16 | $0.2111 \pm 0.0749$ | $0.6732 \pm 0.1449$ |
| 48 | 24 | $0.2014 \pm 0.0829$ | $0.7379 \pm 0.0560$ |
| 56 | 28 | $0.1967 \pm 0.0884$ | $0.7172 \pm 0.0446$ |
| 64 | 32 | $0.1789 \pm 0.0761$ | $0.6937 \pm 0.0480$ |
| 96 | 48 | $0.2116 \pm 0.0885$ | $0.7501 \pm 0.0710$ |
| 128 | 64 | $0.2153 \pm 0.0857$ | $0.7614 \pm 0.0676$ |

First, the window size ($w$) and step size ($\delta$) for the overlapping segments were selected via a grid search on a validation set. The objective of this search was to identify the parameters that maximized the aggregate faithfulness score and robustness score (the sum of Faithfulness@8, @20, @30, and robustness score). Note that the optimal $w$ can offer valuable insights into the temporal scale at which the model identifies important patterns, thus supporting human understanding of model behaviors.

Second, the number of top-h segments used in our cross-view refinement is determined by a ratio, $ratio_h$ of the total number of segments, such that $h = ratio_h \cdot N_S$. As stated in Sec. E.7, we used a fixed ratio of $ratio_h = 0.2$ (20%) for all datasets. This choice is based on the common observation in time series analysis that a relatively small subset of features (around 20%) often accounts for the majority of a model's predictive signal.

Finally, the parameter $\kappa$ in our $KAUC\tilde{S}_{top}$ criterion defines the top fraction of features to prioritize during the quality evaluation of an explanation. While a value of 1.0 would consider all features, this would reduce the focus on the most critical "keystone" features and increase computational overhead. We selected $\kappa = 0.3$, as noted in Appendix D.7, as a principled choice to ensure that the most influential features (to 20%) are studied while remaining computationally efficient.

Table 10: Hyperparameter sensitivity for window and step size on the TwoPatterns dataset. Faithfulness is measured at k=30 (Faithfulness@30).

| Window Size | Step Size | Faithfulness@30 | Robustness |
|---|---|---|---|
| 8 | 4 | $0.7438 \pm 0.0484$ | $0.8608 \pm 0.0532$ |
| 16 | 8 | $0.7508 \pm 0.0587$ | $0.8739 \pm 0.1043$ |
| 32 | 16 | $0.7224 \pm 0.0130$ | $0.9337 \pm 0.0349$ |
| 48 | 24 | $0.7150 \pm 0.0200$ | $0.9716 \pm 0.0253$ |
| 56 | 28 | $0.7214 \pm 0.0123$ | $0.9907 \pm 0.0161$ |
| 64 | 32 | $0.7150 \pm 0.0200$ | $0.9934 \pm 0.0114$ |

Table 11: Sensitivity analysis for the top segment ratio ($h$) and the Keystone AUC fraction ($\kappa$) on the FordA dataset. The best performance for each metric is highlighted in bold.

| Ratio $h$ | $\kappa$ | Faithfulness@30 | Robustness |
|---|---|---|---|
| 0.1 | 0.2 | **0.3266** | 0.6304 |
| | 0.3 | 0.2819 | 0.6803 |
| | 0.5 | 0.2819 | 0.6846 |
| | 1.0 | 0.2819 | 0.6830 |
| 0.2 | 0.2 | 0.3282 | 0.6593 |
| | 0.3 | 0.2880 | 0.6951 |
| | 0.5 | 0.2881 | 0.6967 |
| | 1.0 | 0.2881 | 0.6957 |
| 0.3 | 0.2 | 0.3238 | 0.6738 |
| | 0.3 | 0.2944 | 0.7093 |
| | 0.5 | 0.2943 | **0.7116** |
| | 1.0 | 0.2943 | 0.7065 |

We provide sensitivity analysis of $w$ to the performance of Faithfulness@30 and Robustness in Tab. 9 for FordA dataset, Tab. 10 for TwoPatterns dataset. Overlapping size is set as $w/2$ for simplicity. We report averaged results over 3 DL architectures. For both Tables, there appears to be a trade-off between faithfulness and robustness wrt. the window size $w$. The data shows that faithfulness (Faithfulness@30) generally decreases as $w$ increases. Conversely, robustness tends to increase with a larger $w$, despite some fluctuations. A potential explanation for this trend is that larger $w$ may create segments that are more semantically meaningful and interpretable to a human user. These larger, more stable segments are likely less sensitive to small, localized noise in the input, which could explain the corresponding increase in the robustness score.

We also provide sensitivity analysis on $ratio_h$ and $\kappa$ in Tab. 11 for FordA dataset, Tab. 12 for TwoPatterns dataset. To isolate the impact of these two hyperparameters, which control the cross-view refinement and the adaptive selector respectively, we fixed the window size at 8 and the step size at 4 for this experiment. For FordA, the result reveals a consistent trend when $ratio_h$ is fixed. As k increases, faithfulness tends to decrease, while robustness generally increases. This may be because larger k hinder the focus on the most critical "keystone" features, leading to a lower faithfulness score. The same general behaviour is observed in TwoPatterns. The impact of varying the top-feature ratio, $ratio_h$ appears to be more dataset-dependent. For FordA, with a fixed $\kappa$, increasing $ratio_h$ results in stable faithfulness and improved robustness. In contrast, for TwoPatterns, increasing $ratio_h$ can lead to a slight decrease in faithfulness. This suggests that the optimal ratio of "keystone" features used for refinement can vary, and a more focused refinement may be more beneficial for certain types of data, while others need a higher ratio.

### G.4 Complexity Analysis

For MIX, assuming the time series length $T$, the number of steps to approximate Integrated Gradients (IGs) $m$, the segment step size $\delta$, the window size $w$ and the number of views $|S_V|$, the complexity of a single model query $C$, and the complexity to construct a view via DWT $C_V$, in Phase 1, the number

Table 12: Sensitivity analysis for the top segment ratio ($h$) and the Keystone AUC fraction ($\kappa$) on the TwoPatterns dataset. The best performance for each metric is highlighted in bold.

| Ratio $h$ | $\kappa$ | Faithfulness@30 | Robustness |
|---|---|---|---|
| | 0.2 | 0.7453 | 0.8632 |
| | 0.3 | **0.7475** | 0.8947 |
| 0.1 | 0.5 | 0.7240 | 0.8891 |
| | 1.0 | 0.7240 | 0.8875 |
| | 0.2 | 0.7248 | 0.8573 |
| | 0.3 | 0.7230 | 0.8894 |
| 0.2 | 0.5 | 0.7230 | 0.8893 |
| | 1.0 | 0.7230 | 0.8863 |
| | 0.2 | 0.7247 | 0.8611 |
| | 0.3 | 0.7214 | 0.9298 |
| 0.3 | 0.5 | 0.7214 | **0.9303** |
| | 1.0 | 0.7214 | 0.9303 |

Table 13: Computational cost (seconds per instance) and performance comparison on the FordA and MixedShapesRegularTrain datasets. The best result for each metric is highlighted in bold.

| Dataset | Metric | InteDisUX | SpectralX | MIX |
|---|---|---|---|---|
| **FordA** | Cost/instance | **0.1395** $\pm 0.0830$ | $31.4600 \pm 16.6579$ | $0.4533 \pm 0.1570$ |
| | Faithfulness@30 | $0.0235 \pm 0.0047$ | $0.0813 \pm 0.0163$ | **0.3254** $\pm 0.1382$ |
| | Robustness | $0.4314 \pm 0.1271$ | $0.2546 \pm 0.0761$ | **0.6380** $\pm 0.1042$ |
| **MixedShape RegularTrain** | Cost/instance | **0.345** $\pm 0.108$ | $37.793 \pm 15.075$ | $0.906 \pm 0.671$ |
| | Faithfulness@30 | $0.0432 \pm 0.0268$ | $0.0475 \pm 0.0378$ | **0.6621** $\pm 0.0758$ |
| | Robustness | $0.4448 \pm 0.3038$ | $0.1753 \pm 0.1268$ | **0.7314** $\pm 0.2679$ |

of model queries is $|S_V| \cdot m$, and the time to aggregate the importance scores over overlapping segments is $T/\delta \cdot w$. The total complexity is thus $|S_V| \cdot (m \cdot (C + C_V) + T/\delta \cdot w)$. In Phase 2, the number of model queries for refinement is the same for each view, while the adaptive selector can add approximately $\kappa/s$ model queries ($s$ is the step of the ratio). Thus, the combined complexity for MIX is $|S_V| \cdot (2 \cdot m \cdot (C + C_V) + T/\delta * \cdot w + C \cdot \kappa/s)$.

For InteDisUX, let $N_S$ be the number of initial segments. The worst-case complexity for its greedy interactive refinement process is $m \cdot C + T + N_S^2 \cdot C$. In this expression, the $m \cdot C$ term corresponds to the IG calculation, while the $N_S^2 \cdot C$ term arises from the iterative merging of segments. The parameter $N_S$ is analogous to the $T/\delta$ term in our framework's complexity analysis.

For SpectralX, which employs a greedy strategy, let $k$ be the number of top features to select, $P$ be the number of masks generated, and $R$ be the number of features unmasked in each mask. To select a single best feature, the method must estimate the insertion/deletion score for all candidate features, a process which requires approximately $P$ model queries. Since the greedy strategy selects $k$ features sequentially, the total number of queries is approximately $P * k$. Consequently, the total complexity is approximately $P \cdot k \cdot C$, where $C$ is the cost of a single model query. It is important to note that the default setting for SpectralX is $P = 2000$. For a full comparison with methods like MIX and InteDisUX which provide an importance score for every feature, $k$ would need to be equal to the total number of features of spectralgram, which can be larger than $T$.

Tab. 13 shows averaged computational costs of MIX, InteDisUX and SpectralX over 3 architectures on NVIDIA RTX 4070 GPU. When explaining across 5 views, MIX has *moderately higher computational costs* (approximately 2.5-3x) than InteDisUX but with *significant improvements* in both *faithfulness and robustness*. Compared to SpectralX, MIX is *far more computationally efficient*.

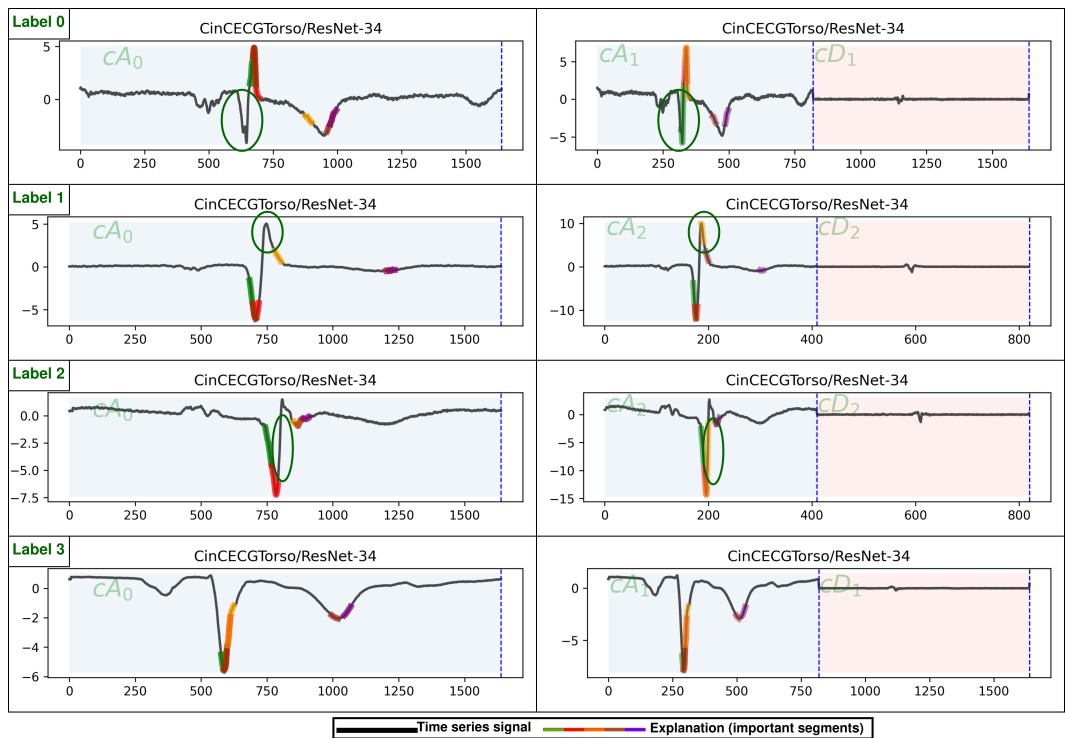

Figure 10: Visualization of the **CinCECGTorso** time series signal (black line) at wavelet level 0 ($cA_0$) and the best wavelet level for explanation for each instance (ranging from level 0 to 5), along with the top explanatory segments (colorful lines, each assigned a distinct color such as green, red, orange, brown, or purple) generated by MIX. Each row represents one instance. The background regions shaded in blue and pink correspond to $cA$ and $cD$ components, respectively, with their names indicated inside each region. The four signals correspond to four different class labels, and differences among these labels can be observed in the visualizations. Key features that are correctly captured by MIX at wavelet levels 1 or 2 ($cA_1$, $cA_2$) but missed at level 0 (raw time series) are indicated with green circles. These results highlight the importance of incorporating multiple DWT views for explanation, demonstrating that a multi-view approach can provide more comprehensive and accurate explanations than relying on the raw time series alone.

## H    Qualitative Results

### H.1    Visualization of Explanations on UCR Dataset

**CinCECGTorso dataset.** We present a visualization of our explanations on the CinCECGTorso dataset and ResNet-34 DL model across multiple instances for each class in Fig. 10. The figure suggests that explanations at wavelet level 1 or  2 tend to capture relevant features more effectively than those at wavelet level 0. Furthermore, the multi-view visualization for a single instance in Fig. 11 indicates that explanation quality is highest at level 3, not at $cA_0$. Taken together, these visualizations highlight the potential value of a multi-view setup for explanation compared to relying solely on the raw time series.

**Wafer dataset.** We present a visualization of our explanations on the Wafer dataset using a ResNet-34 deep learning model to illustrate how explanations can improve when generated from different views (see Fig. 12). These results suggest that employing a multi-view setup in the wavelet domain may enhance the interpretability of deep learning models for time series classification. Furthermore, we observe that each view can both miss and capture important features that are not detected by the other, indicating that cooperation between views is beneficial to provide better explanations.

**TwoPatterns dataset.** We present a visualization of our explanations on the TwoPatterns dataset and ResNet-34 DL model in Fig. 13 to compare explanations at level 0 versus level 1, and in Fig. 14 for

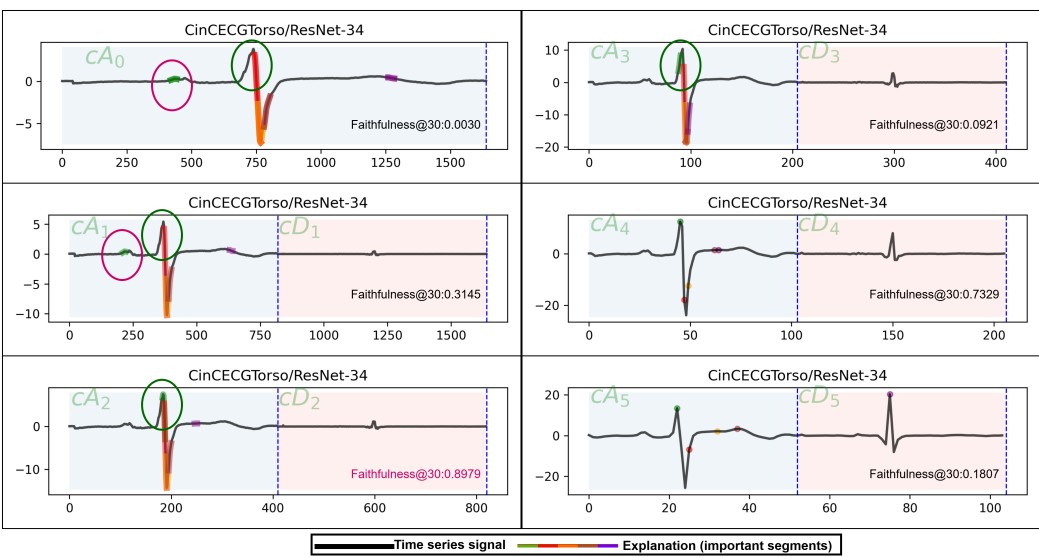

Figure 11: Visualization of the **CinCECGTorso** time series signal (black line) at wavelet level 0 ($cA_0$) and higher levels ($cA_1$ through $cA_5$), together with their corresponding explanations generated by MIX. All figures correspond to a single instance across five wavelet levels. For each level $l$, $cA_l$ and $cD_l$ are shown as two regions with light blue and pink backgrounds, labeled accordingly. The black line represents the time series signal for each component, while the colorful lines indicate the top segments identified by MIX, with each segment assigned a distinct color, including green, red, orange, brown, and purple. Key regions that are missed at levels 0 and 1 but captured at higher levels are marked with green circles. Notably, some important segments are successfully identified in $cA_2$ and $cA_3$ but not in $cA_0$, which is consistent with the Faithfulness@30 evaluation (highest at level 3 and lowest at level 0) shown in each subfigure. At higher levels ($cA_4$, $cA_5$), explanations may also miss crucial segments and are generally less effective than those at $cA_3$. Additionally, $cA_0$ sometimes captures noise (highlighted with a pink circle), which is eliminated at level 3 and above. These results further highlight the importance of a multi-view explanation setup.

level 2. The results indicate that MIX produces better explanations at levels 1 compared to level 0 and level 2 and level 0 can support each other. This finding further underscores the importance of a multi-view setup, consistent with our observations on the CinCECGTorso and Wafer datasets.

## H.2 Visualization of Explanations on Synthetic Dataset

We also provide a visualization of explanations on the synthetic dataset. As shown in Fig. 15 and Fig. 16, MIX effectively captures important features in $cD_1$, which aligns with the way the data was synthesized.

## H.3 Visualization of Explanations on MIT-BIH Dataset

For real-world applications, we visualize our explanations on the MIT-BIH dataset. As shown in Fig. 17, MIX captures important features more effectively by utilizing wavelet levels 2 and 3, rather than relying solely on the raw time series. This leads to a natural question: is it sufficient to search for a single optimal explanation, or is a more flexible, greedy strategy across multiple views preferable? Fig. 18 provides further insight, depending on the instance, explanations at $cA_3$ may capture important features those are missed at $cA_4$, and vice versa. These observations reinforce the importance of a multi-view setup and highlight the value of a greedy selection strategy that traverses multiple DWT levels to produce more faithful and robust explanations. Notably, this finding is consistent with our observations on the Wafer and TwoPatterns datasets.

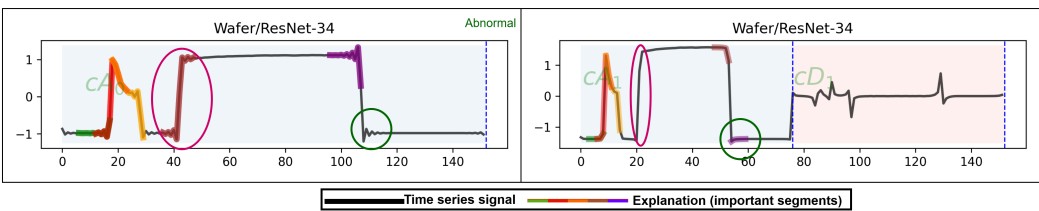

Figure 12: Visualization of the **Wafer** time series signal (black line) with top segments from MIX's explanation represented as colorful lines, each color corresponding to a different top segment, for the two classes: Normal and Abnormal. Each row corresponds to one instance. For each level $l$, $cA_l$ and $cD_l$ are shown as two regions with light blue and pink backgrounds, labeled accordingly. At level 0, MIX can capture features associated with the Abnormal class; however, a small segment (highlighted with a green circle) is missed at this level but is successfully identified in $cA_1$ (right figure). Conversely, some important segments captured in $cA_0$ (highlighted with a pink circle) are not identified at $cA_1$. These results indicate that different views can complement each other to achieve more faithful explanations, supporting the effectiveness of our multi-view setup for time series explanation.

# I Post Hoc vs. Ante Hoc and Local vs. Global Explanation Discussion

While developing effective interpretable-by-design models is a vital direction, we argue that post-hoc explanation remains a crucial and growing area of study. As new increasingly complex architectures like Transformers continue to evolve and achieve SOTA performance on large, complex time series datasets, post-hoc methods remain an essential and practical tool for understanding their behaviour. Moreover, the recent trend towards large, pre-trained foundation models for time series analysis also strengthen the necessity of post-hoc methods as studied in [45]. Even recent hybrid models that combine interpretable and non-interpretable components, such as [90], still require post-hoc analysis to understand their black-box parts.

Table 14: Performance comparison between local and global MIX variants. Results are reported as mean $\pm$ standard deviation.

| Metric | MIX Local | MIX Global |
|---|---|---|
| Faithfulness@8 | $0.7308 \pm 0.0230$ | $0.502 \pm 0.087$ |
| Faithfulness@20 | $0.7756 \pm 0.0794$ | $0.693 \pm 0.016$ |
| Faithfulness@30 | $0.7438 \pm 0.0484$ | $0.723 \pm 0.008$ |
| AUCStop | $0.707 \pm 0.015$ | $0.683 \pm 0.032$ |
| F1S | $0.347 \pm 0.024$ | $0.339 \pm 0.035$ |

Regarding the local versus global scope, our paper focused on local explanations. We have now developed and tested an extension to the MIX framework for generating global explanations. Our approach involves aggregating the local attribution maps from all instances to produce a single, summary map. We evaluated its faithfulness on TwoPatterns dataset and found that it is comparable to local settings, as shown in Table 14.

# J Multi-view vs. Cross-modal Discussion

Our cross-view refinement can be considered as feature fusion of the model. The refinement process works by first identifying "keystone features" from the most faithful view and then using them to guide the attribution calculation in other views via our Keystone-first IG (KIGV) method. This allows strong signals from one perspective to improve the explanations of another. While our current work focuses on different time-frequency views of the same single-modality data. Our current work operates on a single modality, which is a time series signal. The "views" we generate via the Haar DWT are different time-frequency perspectives of this same signal, allowing us to analyze its characteristics at various resolutions. However, we believe "cross-modal integration" is a fantastic concept for future work, especially in a multivariate context. This concept becomes even more

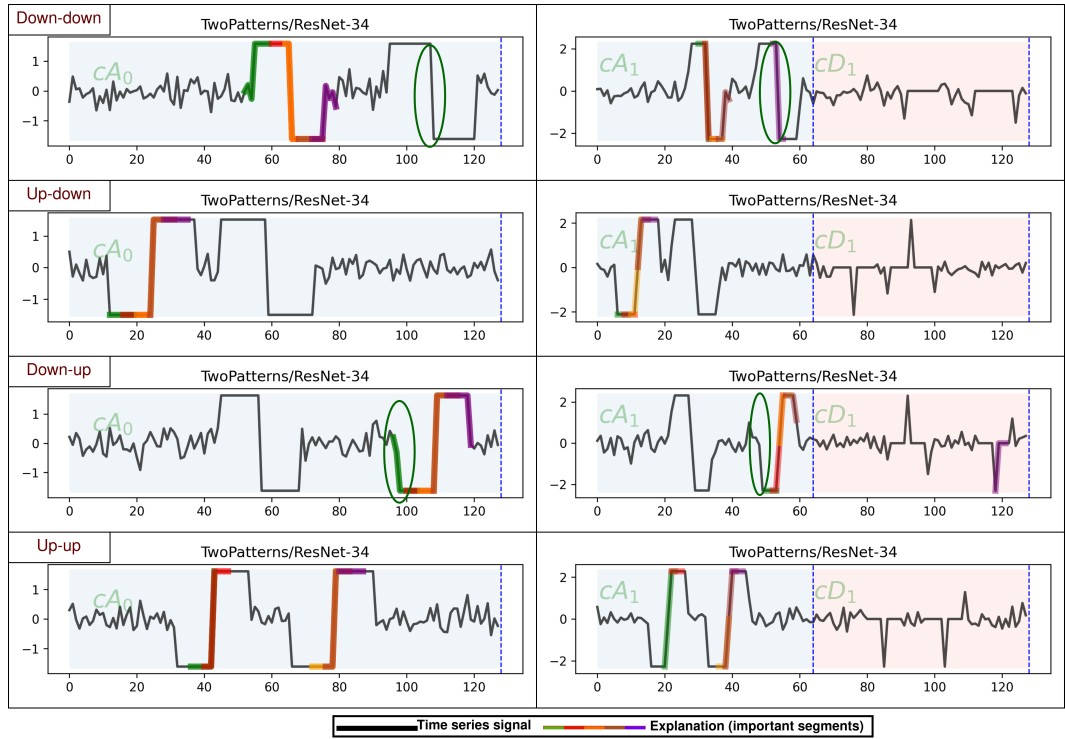

Figure 13: Visualization of the **TwoPatterns** time series signal (black line) with top segments from MIX's explanation represented as colorful lines, each color corresponding to a different top segment, for the four classes represented by two patterns: down-down, up-down, down-up, and up-up. Each row corresponds to one instance across two wavelet levels, 0 and 1. For each level $l$, $cA_l$ and $cD_l$ are shown as two regions with light blue and pink backgrounds, labeled accordingly. For the "down-down" class, MIX fails to capture the segment corresponding to the 2nd "down" pattern in $cA_0$, but the explanation in $cA_1$ successfully highlights this segment as important. Similarly, for the "down-up" class, the segment for the "down" pattern has a redundant part recognized as important segment (marked in green circle) in $cA_0$, but it is not captured in $cA_1$. These findings indicate the importance of a multi-view setup for time series classification (TSC) explanation.

powerful in a multivariate context, where different channels could be treated as distinct but related modalities.

## K Limitations

Even though our framework MIX can provide a good explanation for TSC, it has some limitations that offer avenues for future work. First, while we have extended MIX to handle multivariate time series, our current approach treats each channel independently during the DWT phase; future work could explore multivariate wavelet transforms to better capture cross-channel dependencies . Additionally, as a segment-based attribution method, MIX is subject to the inherent limitation that assigning a single score to a segment can obscure the specific contributions of individual time points within that region .

Furthermore, our framework is designed to provide local, post-hoc explanations. Although some works stated about weakness of post-hoc explanation [79], this approach is particularly relevant in the current landscape, where the rapid development of large-scale foundation models for time series necessitates model-agnostic tools for interpretability [45]. While this local focus offers detailed, instance-specific insights, its primary drawback is the need for users to analyze model behavior on a case-by-case basis. A global explanation framework, which would capture features representative of an entire dataset, is therefore a valuable complementary approach. As stated in other work [67],

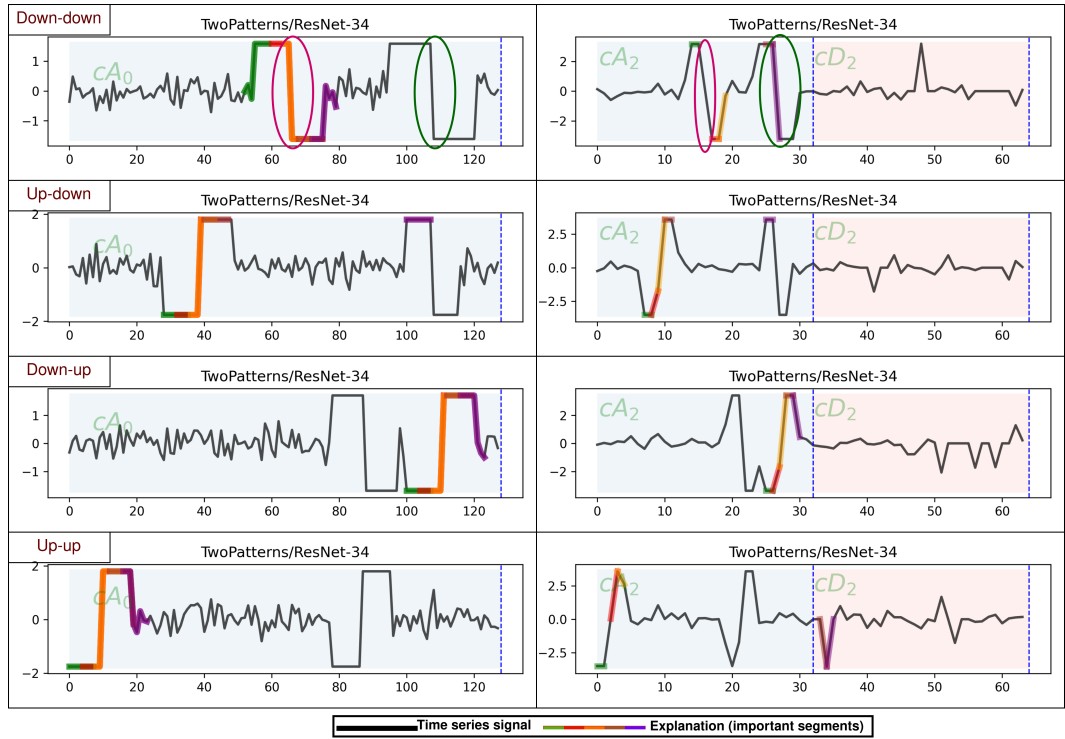

Figure 14: Visualization of the **TwoPatterns** time series signal (black line) with top segments from MIX's explanation represented as colorful lines, each color corresponding to a different top segment, for the four classes defined by two patterns: down-down, up-down, down-up, and up-up. Each row corresponds to one instance across two wavelet levels (0 and 2). For each level $l$, $cA_l$ and $cD_l$ are shown as two regions with light blue and pink backgrounds, labeled accordingly. For the "down-down" class, MIX fails to capture the segment corresponding to the "down" pattern in $cA_0$, but the explanation in $cA_2$ successfully highlights this segment as important. Conversely, an important segment highlighted by a pink circle is recognized by MIX in $cA_0$ but not in $cA_2$. These observations indicate that each view can reveal different important features and can support each other, supporting our motivation for a multi-view explanation setup.

offering both local and global perspectives would provide users with a more comprehensive suite of tools for model understanding.

Finally, we acknowledge the limitations of the current evaluation paradigm for local XAI, which relies heavily on faithfulness scores from ablation studies, particularly the practice of removing segments to evaluate time series explanations. We recognize that such metrics may be insufficient to definitively identify a single "most effective explanatory perspective". This points to a broader challenge in the field to develop more holistic evaluation frameworks, as noted in other research, as paper [67] stated.

Besides, our MIX framework was designed with an assumption about "no perfect metric" in mind. Hence it is not rigidly tied to a specific metric. The framework's key strength is its flexibility. The selection criterion used to identify the "best view" is modular. As the XAI field matures and develops better and more holistic evaluation metrics, they can be easily integrated into our framework to replace the current faithfulness average. The adaptive selector in Phase 2, which acts as a quality gate, is a key example of this modular and principled design. In addition, we can combine different metrics to have a better evaluation (by covering wider aspects) like the way we did in our paper with 3 metrics. This is inspired from the way different classification evaluation metrics e.g. Acc, F1, AUC are combined in machine learning classification problems.

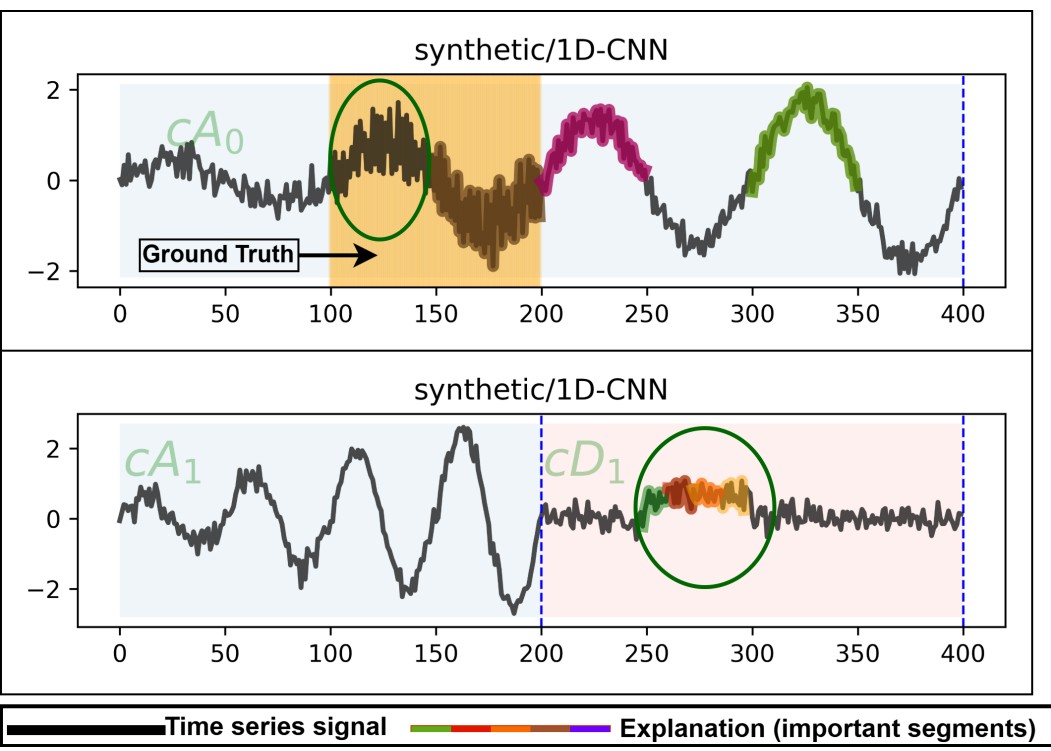

Figure 15: Visualization on the **synthetic** dataset with the top 100 important features highlighted by selecting top segments. All figures corresponds to one instance across two wavelet levels ( 0 and 1). For each level $l$, $cA_l$ and $cD_l$ are shown as two regions with light blue and pink backgrounds, labeled accordingly. The ground truth region for this dataset is from time step 100 to 200 in $cA_0$, corresponding to steps 250 to 300 in $cD_1$ based on the synthetic process, highlighted by color yellow. MIX successfully captures the key features in $cD_1$, correctly identifying the important region marked in green circle (positions 250–300 in $cD_1$, which maps to 100–200 in $cA_0$ and matches the ground truth). In contrast, the explanation on $cA_0$ alone misses part of the ground truth (notably, the segment from approximately time step 100 to 150 remains unhighlighted) marked in green circle, demonstrating the benefit of multi-view analysis.

## L  Broader Impacts

Our work proposes a novel explanation for TSC, which can apply to various applications such as ECG anomaly detection, sensor data analysis, and help people in those fields develop deep learning and explain them. With a good explanation framework, deep learning models will become more trustworthy for humans and motivate users to integrate models into their systems. However, post-hoc explanation can make users understand which features are influential in the model's decision, but ignore causality within them. It is necessary to integrate causality and reasoning to post-hoc explanation, but it is out of the scope of our studies. Also, our research does not directly introduce societal or environmental risks. However, as the underlying models remain black-box in nature, the method should not be viewed as a substitute for rigorous model validation or fairness analysis.

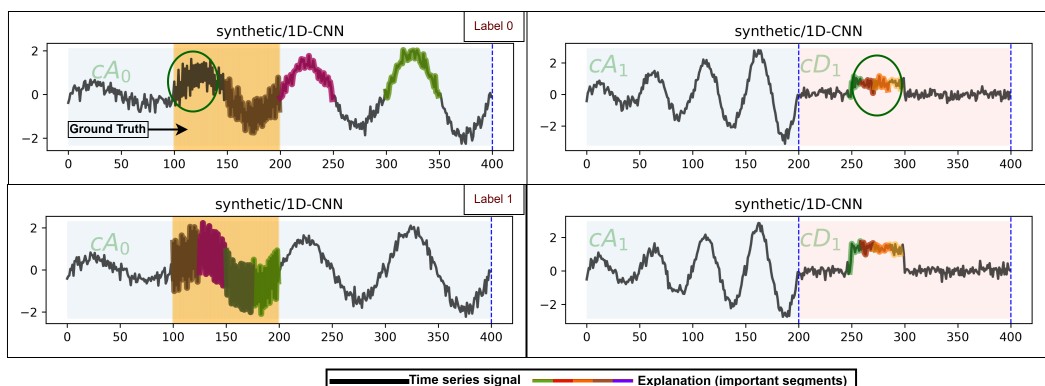

Figure 16: Visualization on the **synthetic** dataset with the top 100 important features highlighted by selecting top segments. Each row corresponds to one instance across two wavelet levels (0 and 1). For each level $l$, $cA_l$ and $cD_l$ are shown as two regions with light blue and pink backgrounds, labeled accordingly. The ground truth region for this dataset is from time step 100 to 200 in $cA_0$, corresponding to steps 250 to 300 in $cD_1$ based on the synthetic process, highlighted by color yellow. MIX successfully captures the key features in $cD_1$, correctly identifying the important region marked with a green circle in both classes (positions 50 to 100 (250–300 as shift from $cA_1$) in $cD_1$, mapping to 100–200 in $cA_0$, matching the ground truth). In contrast, the explanation on $cA_0$ alone misses part of the ground truth in class 0 (notably, the segment from approximately time steps 100 to 150 remains unhighlighted, as marked by the green circle), indicating the benefit of multi-view analysis.

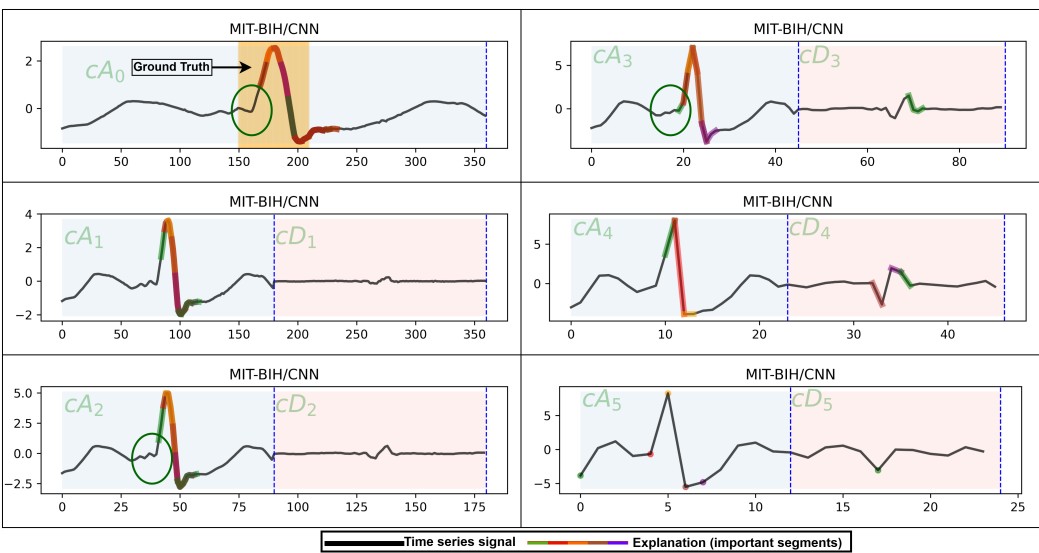

Figure 17: Visualization on the **MIT-BIH** dataset with the top segments highlighted. All figures represent a single instance across five wavelet levels. The ground truth, as annotated by cardiologists according to [66], typically corresponds to the peak region in the middle of the signal, highlighted by color yellow. For each level $l$, $cA_l$ and $cD_l$ are shown as two regions with light blue and pink backgrounds, labeled accordingly. Explanations using only $cA_0$ miss some important features, whereas $cA_2$, and $cA_3$ more effectively capture the relevant segments, which are marked with green circles.

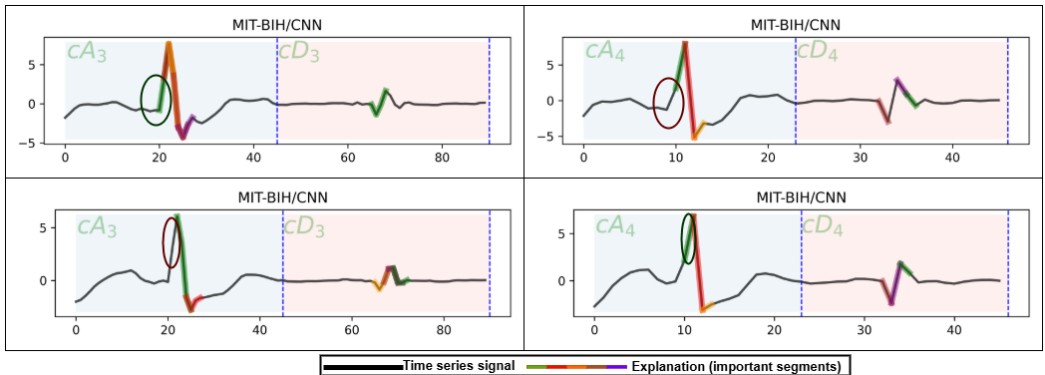

Figure 18: Visualization on the **MIT-BIH** dataset with the top segments highlighted. Each row shows one instance at two DWT levels (3 and 4). For each level $l$, $cA_l$ and $cD_l$ are shown as two regions with light blue and pink backgrounds, labeled accordingly. In the instance in the first row, the explanation in $cA_3$ at level 3 covers important features more effectively than $cA_4$ at level 4. The region captured by $cA_3$ is indicated with a green circle, while the corresponding region in $cA_4$ (marked with a red circle) is not highlighted. Conversely, for the instance in the second row, MIX at level 4 with $cA_4$ highlights important features that are missed at level 3 with $cA_3$ (with missed regions marked by red circles in $cA_3$ and by green circles in $cA_4$ where they are correctly identified). These observations suggest that a multi-view setup and cross-view aggregation are beneficial for achieving more faithful explanations.

