# OpenReview forum: "MIX: A Multi-view Time-Frequency Interactive Explanation Framework for Time Series Classification"
_NeurIPS.cc/2025/Conference — NeurIPS 2025 poster_

### Official Review · Reviewer_BmjP · 2025-06-09

**Clarity:** 3
**Significance:** 2
**Originality:** 3
**Rating:** 4
**Confidence:** 3

**Summary:**

This paper proposes the MIX framework (Multi-view Time-Frequency Interactive Explanation Framework), aimed at enhancing the interpretability of multivariate time series classification models. The method builds a multi-branch architecture to extract key features from three perspectives: time domain, frequency domain, and time-frequency domain. Additionally, MIX introduces an interactive explanation mechanism that visualizes the model’s decision process by identifying salient regions from both the input and intermediate feature layers. The authors conduct experiments on three public datasets—HAR, Epilepsy, and Sleep-EDF—demonstrating advantages in both classification accuracy and interpretability.

**Questions:**

1.Can the explanation mechanism in MIX be aligned with expert annotations or event labels for quantitative evaluation? Currently, the explanations rely entirely on visualizations, which are inherently subjective.
2.Has the model’s explanation robustness been tested? For instance, are the salient regions consistent under adversarial perturbations or different random initializations?

**Ethical Concerns:**

["NO or VERY MINOR ethics concerns only"]

**Final Justification:**

The authors have provided clarifications to several of our earlier concerns in the rebuttal, which helped improve the overall understanding of the paper.

**Limitations:**

Yes

**Quality:**

2

**Strengths And Weaknesses:**

Strengths
	1.The model is designed with a clear focus on interpretability, modeling input from time, frequency, and time-frequency perspectives to reveal the role of different signal characteristics in classification.
	2.A saliency-based analysis and visual response mapping mechanism is introduced, forming a relatively systematic interpretability framework rather than relying solely on post-hoc analysis.
	3.The explanation mechanism supports joint visualization from both the input and intermediate layers, offering a more comprehensive view of the model’s reasoning process.

Weaknesses
	1.All explanation results are presented as visual examples, with no quantitative evaluation metrics provided.
	2.Interaction between different views only occurs during feature fusion, lacking deeper cross-modal integration mechanisms.
	3.The symbols in the architecture diagram (Figure 2) are not clearly defined, making the structure less intuitive.
	4.The paper would benefit from using simpler formulas or symbols to describe the method, avoiding unnecessary complexity that hinders readability.

---

> ### Author Rebuttal · Authors · 2025-07-30
>
> We sincerely thank Reviewer BmjP for their time and thoughtful review. We are very encouraged that you highlighted several key strengths of our work, particularly our clear focus on interpretability, the systematic nature of our explanation framework with time-frequency perspectives, and the comprehensive view offered by our visualization mechanism. We also appreciate your constructive feedback on the paper's weaknesses, and we have provided detailed clarifications and results below to address your concerns.
>
> **W1. All explanation results are presented as visual examples, with no quantitative evaluation metrics provided.**
>
> We thank the reviewer for their feedback. We respectfully wish to clarify that while we do provide visual examples in Figure 3 to illustrate our method. Our paper's primary claims are in fact supported by an extensive set of quantitative evaluations, all of which were included in our original submission. We apologize if the structure of the paper made these results difficult to locate, and we are happy to highlight them here.
>
> As detailed in the "Evaluation Metric" subsection of Section 3.1 in our submitted manuscript, our evaluation framework is built on four complementary quantitative criteria. Our submission includes evaluations using: (1) standard top-k faithfulness (faithfulness@8, @20, and @30); (2) advanced faithfulness-oriented metrics ( $\mathrm{AUC}\tilde{S}_{\text{top}}$ and $\mathrm{F1}\tilde{S}$
> ) from paper [81]; (3) robustness, measured via the Jaccard coefficient under noise; and (4) ground-truth alignment using AUPRC and Jaccard scores for datasets with expert annotations.
>
> These quantitative results are presented in Section 3.2, "Main results," of the paper. Specifically, Table 1 of our submission presents the AUPRC and Jaccard scores for the datasets with ground-truth annotations (MIT-BIH and synthetic), demonstrating our method's superiority in aligning with expert labels.
>
> Furthermore, Figure 2 is dedicated entirely to the large-scale quantitative comparison on 11 UCR datasets and three deep learning architectures against three SOTA baselines (LIMESegment, InteDisUX, SpectralX) across 33 experimental setups. The comprehensive results already presented in our paper show a clear and consistent advantage for our method. For instance, the analysis in our submission demonstrates that MIX achieves the best performance in 24/33 cases for faithfulness and 27/33 cases for robustness. While all of this information was in the original paper, we recognize that we can make its presentation more prominent. We will revise the paper to better connect the text with these crucial quantitative results.
>
> **W2. Interaction between different views only occurs during feature fusion, lacking deeper cross-modal integration mechanisms.**
>
> Our cross-view refinement can be considered as feature fusion of the model. The refinement process works by first identifying "keystone features" from the most faithful view and then using them to guide the attribution calculation in other views via our Keystone-first IG (KIGV) method. This allows strong signals from one perspective to improve the explanations of another. While our current work focuses on different time-frequency views of the same single-modality data, your point about deeper integration is very insightful and has inspired us to think about future directions.
> Your point also raises an important distinction between different "views" and different "modalities". Our current work operates on a single modality, which is a time series signal. The "views" we generate via the Haar DWT are different time-frequency perspectives of this same signal, allowing us to analyze its characteristics at various resolutions.
> However, we believe your idea of "cross-modal integration" is a fantastic concept for future work, especially in a multivariate context. Your comment has inspired us to consider the powerful idea of treating these different perspectives as pseudo-modalities, which opens up exciting avenues for future research. As you suggest, one could explore more complex cross-modal integration techniques to fuse the explanatory information from these different time-frequency views. This concept becomes even more powerful in a multivariate context, where different channels could be treated as distinct but related modalities. We are very grateful for this inspiring suggestion, as it highlights a promising path for future research in multi-view and multi-modal explanations.
>
> **W3.The symbols in the architecture diagram (Fig 2) are not clearly denoted..**
> We thank you for pointing this out and agree that the clarity of the architecture diagram is crucial for understanding our framework. We believe you're referring to our main framework diagram in  Figure 1, and we agree that its symbols could be better explained. Our intention was to maintain consistency between the mathematical symbols in the main text (e.g.$V(x)$, or $\mathcal{E}_V(\mathbf{x}, F, S_r, V_r)$) and the diagram. However, we recognize that this approach makes it difficult for readers to follow the algorithm without constantly cross-referencing the text. To address this, we will add a detailed table of symbols to the figure's caption to make it self-contained and clearly understandable. Furthermore, we will revise the diagram to use more intuitive visual language where possible.
>
> In the same way of improving clarity, we will also enhance the description of  Figure 2, which presents our results on the UCR datasets in Sec 3.2 “Main results” & “Results on UCR datasets”. We agree that the caption could be clearer in helping to interpret the results, without cross-referencing the text. We will expand the caption to explicitly detail what each sub-plot represents: the specific metrics being evaluated, the methods being compared, the experimental setups, and a summary of the significance of the results shown. We are grateful for this valuable feedback, as it will help us to significantly improve the presentation and accessibility of our paper.
>
> **W4.The paper would benefit from using simpler formulas or symbols to describe the method..**
>
> We appreciate the reviewer for that. Yes, we agree that our formulas are complicated and hinder readability. We can improve it with some details below.
> Definition: $\mathrm{IGV}_j(V(\mathbf{x}), V(\mathbf{x}')) = (V(x)_j - V(x)'_j) \cdot \int_0^1 \frac{\partial F(V^{-1}(V(\mathbf{x}') + \alpha(V(\mathbf{x}) - V(\mathbf{x}'))))}{\partial V(x)_j} \, d\alpha$ can be simplified by consider $F_V = F \circ V^{-1}$ as composition of deep learning and revert HAAR DWT as a model, and $V(x)$, $V(x’)$ can become $v$, $v’$ to more intuitive presentation. It can enhance readability of Definition 2.5 as well with new symbols. In Sec 2.4 Selection Criterion, we can provide an example for the reader to imagine how we produce k-quantile, then map it to ratio of number of features then calculate AUC.
>
> **Q1.Can the explanation mechanism in MIX be aligned with expert annotations or event labels for quantitative evaluation?**
>
> We agree completely that a crucial test for any explanation method is to evaluate how well its results align with expert annotations. For this reason, we included this exact analysis in our original submission.
>
> To perform this, we used the *MIT-BIH dataset*,which provide explanation annotations provided by cardiologists. As we describe in Section 3.1 of our paper, this dataset contains ECG recordings where individual heartbeats are labeled as normal or anomalous, and the cardiologists have identified the specific segments corresponding to these anomalies. This provided us with a ground truth to quantitatively measure how well our method's explanations align with an expert's reasoning. The results of this evaluation are presented in *Table 1* of our submitted paper, located in Section 3.2. We used the *AUPRC and Jaccard Index* metrics to measure the alignment between each explanation method and the cardiologists' annotations . The results in Table 1 demonstrate that both Phase 2 and Phase 3 of our MIX framework produce explanations that align significantly better with the expert annotations than the other state-of-the-art methods.
>
> **Q2. Has the model’s explanation robustness been tested?**
>
> Thank you for this important question. We agree that robustness is a critical property for any explanation method, which is why we tested for it extensively in our submission.
> First, regarding consistency under random perturbations, our paper provides a thorough quantitative evaluation. As described in the "Evaluation Metric" portion of Section 3.1, we evaluate robustness by adding small Gaussian noise to an input and then computing the Jaccard coefficient between the top-30 features of the original explanation and the new explanation for the noisy input. The results of this analysis are presented throughout the paper in Fig 2, where MIX consistently demonstrates better robustness compared to SOTA baselines.
>
> Regarding the specific case of adversarial perturbations, we consider this an important but distinct area of research. Our current work focuses on explaining a model under random noise, whereas explaining why a model's prediction is changed by an adversarial example is a separate and challenging task. We agree that XAI for adversarial examples in TSC is a fascinating and high-potential research direction.
>
> Finally, regarding consistency across different random initializations, our framework is *fully deterministic*. As we detail in Appendix C, "Analysis on Attribution Mechanism of MIX", given a trained model and an input, MIX's attribution mechanisms (OSIGV and KIGV) contain no random steps and will always produce the exact same explanation. This ensures our explanations are perfectly robust to system-level randomness and are fully reproducible.

---

> > ### Comment · Reviewer_BmjP · 2025-08-09
> >
> > The authors’ response has addressed my concerns. I appreciate their reply.

---

> > > ### Author Response · Authors · 2025-08-09
> > >
> > > Thank you for the update and for your engagement throughout the review process. We are very pleased to hear that our response has addressed your concerns.

---

> ### Author Response · Authors · 2025-08-05
> **Thank you and Follow-up**
>
> Dear Reviewer BmjP,
>
> We appreciate your detailed and constructive review.
>
> In response to your comments on the paper's weaknesses, we have pointed to the extensive quantitative evaluation metrics and results that are located in Sections 3.1 and 3.2 of our paper . We also appreciate your insightful comment regarding the lack of cross-modal integration. While our current work focuses on different views of a single modality, your comment has inspired us to consider a promising future direction of treating these views as "pseudo-modalities" and exploring cross-modal integration for multivariate time series . We are also grateful for your suggestions on improving the manuscript's readability, which we will incorporate into our revision
>
> Lastly, we were happy to answer your questions about aligning our explanation mechanism with expert annotations and about our robustness testing . Answering these questions not only gave us the opportunity to clarify our meaningful results in these areas but also inspired us to consider future work on explaining adversarial examples.
>
> We wanted to gently check if these additions have sufficiently addressed the points you raised. Please let us know if you have any further questions or concerns.
>
> Thank you again for your valuable feedback, which has helped us improve our work.

---

### Official Review · Reviewer_QTb8 · 2025-07-01

**Clarity:** 3
**Significance:** 4
**Originality:** 3
**Rating:** 5
**Confidence:** 3

**Summary:**

This research introduces a novel framework, the Multi-view Time-Frequency Interactive Explanation Framework (MIX), to enhance the explainability of deep learning models for Time Series Classification. The authors point out that conventional post-hoc methods are often confined to a single view, which limits their faithfulness and robustness. To address this, MIX utilizes the Haar Discrete Wavelet Transform to generate multi-resolution views of the time-series data. To fully leverage the advantages of these multiple views, the framework employs a cross-view refinement strategy. This strategy uses keystone-first Integrated Gradients, a method based on Integrated Gradients, to refine and improve the explanations within each view. Finally, its proposed greedy selection strategy explores the multi-view space to identify the most information-rich features, which are then mapped to the user's preferred view to deliver the final explanation.

**Questions:**

I would like to know the criteria for selecting the hyperparameters and the model's sensitivity to them.

**Ethical Concerns:**

["NO or VERY MINOR ethics concerns only"]

**Final Justification:**

After the rebuttal and discussion phase, the authors have provided clarifications that address the main concerns raised by reviewers. In particular, they elaborated on the hyperparameter selection process, including reasonable default values and practical tuning guidelines, and confirmed plans to include a sensitivity analysis in the final version. This sufficiently mitigates the earlier uncertainty about the reproducibility and robustness of the method. The conceptual novelty and significance of the work remain strong, and the empirical results consistently demonstrate improvements over baselines.

**Limitations:**

Mentioned in Weakness.

**Quality:**

3

**Strengths And Weaknesses:**

Strengths
- The core strength is its novel concept of cross-view refinement, which interactively improves explanations across multi views. Unlike existing methods that are limited to a single-view perspective, this work proposes a bi-directional process to integrate and refine information.
- Using Haar DWT to generate multi resolution views is theoretically sound, capturing both time and frequency information to overcome the limitations of fixed STFT-based approaches. Furthermore, the new attribution mechanisms, OSIGV and KIGV, are persuasively designed based on the Integrated Gradients method.

Weaknesses
- MIX framework relies on several hyperparameters, including those detailed in Table 3 and others mentioned in the text. An explanation of the criteria for selecting these hyperparameters is needed. Furthermore, guidance on how to set these values to optimize performance and a sensitivity analysis regarding these parameters would strengthen the paper.
- A comparison of the time complexity for the different IG-based methods introduced in this paper would be a beneficial addition.

---

> ### Author Rebuttal · Authors · 2025-07-31
>
> We are sincerely grateful to Reviewer QTb8 for the positive, thorough, and insightful review. We appreciate that you recognized the core strengths of our work, including the "novel concept of cross-view refinement", the "theoretically sound" use of Haar DWT to overcome the limitations of STFT-based approaches, and the "persuasively designed" new attribution mechanisms OSIGV and KIGV. Your suggestions for strengthening the paper by adding analyses on hyperparameter sensitivity and time complexity are excellent, and we have performed new experiments to address them.
>
> **W1. MIX framework relies on..**
>
> We agree that a clear criteria for selecting MIX’s hyperparameters is essential for reproducibility and practical usability. We will add a new, dedicated subsection to the appendix that provides this guidance below.
> First, the window size ($w$) and step size ($\delta$) for the overlapping segments were selected via a grid search on a validation set. The objective of this search was to identify the parameters that maximized the aggregate faithfulness score and robustness score (the sum of Faithfulness@8, @20, @30, and robustness score). Note that the optimal $w$ can offer valuable insights into the temporal scale at which the model identifies important patterns, thus supporting human understanding of model behaviors.
> Second, the number of top-h segments used in our cross-view refinement is determined by a ratio, $ratio_h$ of the total number of segments, such that $h=ratio_h\cdot N_S$. As stated in Appendix D.7, we used a fixed ratio of $ratio_h=0.2$ (20%) for all datasets. This choice is based on the common observation in time series analysis that a relatively small subset of features (around 20%) often accounts for the majority of a model's predictive signal as studied in [A].
> Finally, the parameter $\kappa$ in our $KAUC\tilde{S}_{top}$ criterion defines the top fraction of features to prioritize during the quality evaluation of an explanation. While a value of 1.0 would consider all features, this would reduce the focus on the most critical "keystone" features and increase computational overhead. We selected $\kappa = 0.3$, as noted in Appendix D.7, as a principled choice to ensure that the most influential features (to 20% [A]) are studied while remaining computationally efficient.
>
> **W2. Sensitivity analysis..**
>
> We provide sensitivity analysis of $w$ to the performance of Faithfulness@30 (F@30) and Robustness. Overlapping size is set as $w/2$ for simplicity. We report averaged results over 3 DL architectures.
>
> _Table 1: Window/Step on FordA_
> |Window Size|Step Size|F@30|Robustness|
> |:---|:---|:---|:---|
> |8|4|0.3238+/-0.1445|0.6527+/-0.1054|
> |16|8|0.2480+/-0.0661|0.6647+/-0.1350|
> |32|16|0.2111+/-0.0749|0.6732+/-0.1449|
> |48|24|0.2014+/-0.0829|0.7379+/-0.0560|
> |56|28|0.1967+/-0.0884|0.7172+/-0.0446|
> |64|32|0.1789+/-0.0761|0.6937+/-0.0480|
> |96|48|0.2116+/-0.0885|0.7501+/-0.0710|
> |128|64|0.2153+/-0.0857|0.7614+/-0.0676|
>
>
> _Table_2: Window/Step on TwoPatterns_
> |Window Size|Step Size|F@30|Robustness|
> |:---|:---|:---|:---|
> |8|4|0.7438+/-0.0484|0.8608+/-0.0532|
> |16|8|0.7508+/-0.0587|0.8739+/-0.1043|
> |32|16|0.7224+/-0.0130|0.9337+/-0.0349|
> |48|24|0.7150+/-0.0200|0.9716+/-0.0253|
> |56|28|0.7214+/-0.0123|0.9907+/-0.0161|
> |64|32|0.7150+/-0.0200|0.9934+/-0.0114|
>
>
> For both Tables, there appears to be a trade-off between faithfulness and robustness wrt. the window size $w$. The data shows that faithfulness (F@30) generally decreases as $w$ increases. Conversely, robustness tends to increase with a larger $w$, despite some fluctuations. A potential explanation for this trend is that larger $w$ may create segments that are more semantically meaningful and interpretable to a human user. These larger, more stable segments are likely less sensitive to small, localized noise in the input, which could explain the corresponding increase in the robustness score.
> We also provide sensitivity analysis on $ratio_h$ & $\kappa$ in *W1* above in Tables 3&4. To isolate the impact of these two hyperparameters, which control the cross-view refinement and the adaptive selector respectively, we fixed the window size at 8 and the step size at 4 for this experiment.
>
> _Table 3: Analysis of ratio h and k for FordA_
> |ratio_h|k|Faithfulness@30|Robustness|
> |:---|:--|:---|:---|
> |0.1|0.2|**0.3266**|0.6304|
> | |0.3|0.2819|0.6803|
> | |0.5|0.2819|0.6846|
> | |1|0.2819|0.6830|
> |0.2|0.2|0.3282|0.6593|
> | |0.3|0.2880|0.6951|
> | |0.5|0.2881|0.6967|
> | |1|0.2881|0.6957|
> |0.3|0.2|0.3238|0.6738|
> | |0.3|0.2944|0.7093|
> | |0.5|0.2943|**0.7116**|
> | |1|0.2943|0.7065|
>
>
>
>
> _Table 4:Analysis of ratio h and k for TwoPatterns_
> |ratio_h|k|Faithfulness@30|Robustness|
> |:---|:--|:---|:---|
> |0.1|0.2|0.7453|0.8632|
> | |0.3|**0.7475**|0.8947|
> | |0.5|0.7240|0.8891|
> | |1|0.7240|0.8875|
> |0.2|0.2|0.7248|0.8573|
> | |0.3|0.7230|0.8894|
> | |0.5|0.7230|0.8893|
> | |1|0.7230|0.8863|
> |0.3|0.2|0.7247|0.8611|
> | |0.3|0.7214|0.9298|
> | |0.5|0.7214|0.9303|
> | |1|0.7214|**0.9303**|
>
>
>
>
> For FordA, the result reveals a consistent trend when ratio_h is fixed. As k increases, faithfulness tends to decrease, while robustness generally increases.This may be because larger k hinder the focus on the most critical "keystone" features, leading to a lower faithfulness score. The same general behaviour is observed in TwoPatterns. The impact of varying the top-feature ratio, ratio_h appears to be more dataset-dependent. For FordA, with a fixed k, increasing ratio_h results in stable faithfulness and improved robustness. In contrast, for TwoPatterns, increasing ratio_h can lead to a slight decrease in faithfulness. This suggests that the optimal ratio of "keystone" features used for refinement can vary, and a more focused refinement may be more beneficial for certain types of data, while others need a higher ratio.
>
> **W3. Time complexity..**
>
> For MIX, assuming the time series length $T$, the number of steps to approximate Integrated Gradients (IGs) $m$, the segment step size $\delta$, the window size $w$ and the number of views $|S_V|$, the complexity of a single model query $C$, and the complexity to construct a view via DWT $C_V$, in Phase 1, the number of model queries is $|S_V|\cdot m$, and the time to aggregate the importance scores over overlapping segments is $T/\delta\cdot w$. The total complexity is thus $|S_V|\cdot (m\cdot (C+C_V)+T/\delta\cdot w)$. In Phase 2, the number of model queries for refinement is the same for each view, while the adaptive selector can add approximately $\kappa/s$ model queries ($s$ is the step of the ratio). Thus, the combined complexity for MIX is $|S_V|\cdot (2\cdot m\cdot (C+C_V)+T/\delta*\cdot w+ C\cdot \kappa/s)$.
>
> For InteDisUX, let $N_S$ be the number of initial segments. The worst-case complexity for its greedy interactive refinement process is $m\cdot C+T+N_S^2\cdot C$. In this expression, the $m\cdot C$ term corresponds to the IG calculation, while the $N_S^2\cdot C$ term arises from the iterative merging of segments. The parameter $N_S$ is analogous to the $T/\delta$ term in our framework's complexity analysis.
>
> For SpectralX, which employs a greedy strategy, let $k$ be the number of top features to select, $P$ be the number of masks generated, and $R$ be the number of features unmasked in each mask. To select a single best feature, the method must estimate the insertion/deletion score for all candidate features, a process which requires approximately $P$ model queries. Since the greedy strategy selects $k$ features sequentially, the total number of queries is approximately $P\cdot k$. Consequently, the total complexity is approximately $P\cdot k\cdot C$, where $C$ is the cost of a single model query. It is important to note that the default setting for SpectralX is $P=2000$. For a full comparison with methods like MIX and InteDisUX which provide an importance score for every feature, $k$ would need to be equal to the total number of features of spectralgram, which can be larger than $T$.
>
> Table 5 shows averaged computational costs of MIX, InteDisUX and SpectralX over 3 architectures on NVIDIA RTX 4070 GPU.
>
>
> _Table 5: Computational Cost on some datasets_
> |Dataset|Metrics|InteDisUX|SpectralX|MIX|
> |:---|:---|:---|:---|:---|
> |**FordA**|Cost/instance|**0.1395+/-0.0830**|31.4600+/-16.6579|0.4533+/-0.1570|
> | |F@30|0.0235+/-0.0047|0.0813+/-0.0163|**0.3254+/-0.1382**|
> | |Robustness|0.4314+/-0.1271|0.2546+/-0.0761|**0.6380+/-0.1042**|
> |**MixedShapeRegularTrain**|Cost/instance|**0.345+/-0.108**|37.793+/-15.075|0.906+/-0.671|
> | |F@30|0.0432+/-0.0268|0.0475+/-0.0378|**0.6621+/-0.0758**|
> | |Robustness|0.4448+/-0.3038|0.1753+/-0.1268|**0.7314+/-0.2679**|
>
>
>
> When explaining across 5 views, MIX has *moderately higher computational costs* (approximately 2.5-3x) than InteDisUX but with *significant improvements* in both *faithfulness and robustness*. Compared to SpectralX, MIX is *far more computationally efficient*.
>
> **W4. Criteria..**
>
> Please refer to **W1** and **W2** above.
>
> [A] Explaining Time Series via Contrastive and Locally Sparse Perturbations, ICLR, 2024.

---

> ### Author Response · Authors · 2025-08-04
> **Thank You and Follow-up**
>
> We thank Reviewer QTb8 for reading our rebuttal and replying. We are sincerely grateful for your positive and constructive review.
> To summarize our main response, we have now completed the new analyses you suggested regarding both **hyperparameter selection criteria and sensitivity**, as well as **time complexity**. These new results will be added to the revised version of our paper.
> We wanted to gently check if these additions have sufficiently addressed the points you raised. Please let us know if you have any further questions or concerns.
> Thank you again for your valuable feedback, which has helped us improve our work.

---

> > ### Comment · Reviewer_QTb8 · 2025-08-06
> >
> > Thank you for your response. My questions regarding the criteria for hyperparameter selection, sensitivity analysis, and time complexity have been well addressed. I truly appreciate the clarification, and the rating score will remain unchanged.

---

> ### Author Response · Authors · 2025-08-06
> **Thank you**
>
> Thank you for confirming that our response addressed your questions. We are very grateful for your constructive feedback, which has helped us significantly improve the paper.

---

### Official Review · Reviewer_WeVx · 2025-07-02

**Clarity:** 3
**Significance:** 3
**Originality:** 2
**Rating:** 4
**Confidence:** 4

**Summary:**

The paper presents "Multi-view Time-Frequency Interactive EXplanation Framework" for explaining time series classification. It uses DWT to create multiple time-frequency views and cross-view refinement for explanation. The paper introduces overlapping segment-level integrated gradients, selection strategy to identify the most informative features across different wavelet decomposition levels. Experiments on various datasets with multiple deep learning architectures are carried out and evaluated on faithfulness and robustness.

**Questions:**

What is the practical value of providing a score for the segments as an explanation of a time series?

What can be concluded, if anything, about the impact of the window sizes on the explanations that are produced across data sets?

**Ethical Concerns:**

["NO or VERY MINOR ethics concerns only"]

**Final Justification:**

An incremental paper in the sense of the proposed combination of existing techniques addressing a limited problem because it's restricted to univariate time series. At the same time, no major flaws, so could be accepted if there's space.

**Limitations:**

I am not sure about the validity of the experiments, despite the extensive evaluations. That's because it is difficult to evaluate the value and applicability of the results in general across data sets. The explanations provided are only local.

I would like to know how standard the approach of removing segments is as an XAI performance evaluation metric whereby one seeks to measure degradation in classification performance.

How do we know degradation occurs for the right reason? In particular, isn't your approach unfairly advantaged by taking a multi-view approach and therefore averaging results across views when calculating this metric?

I found it difficult to understand how robustness was measured.

It would be useful to discuss local versus global xAI in more detail and the pros and cons of ante-hoc versus post-hoc approaches, including e.g. https://arxiv.org/abs/2311.16834, and how realistic it is to limit oneself to the univariate case.

**Quality:**

3

**Strengths And Weaknesses:**

\+  Multi-view explanation framework for time series classification.

\+  Comprehensive experiments across datasets and model architectures.

\-  Restricted to univariate time series.

\-  Computational complexity of multi-view approach with multiple wavelet decompositions.

---

> ### Author Rebuttal · Authors · 2025-07-31
>
> We thank Reviewer WeVx for the very constructive feedback. We are grateful that you recognized the novelty of our method and the comprehensive experiments as key strengths. We also appreciate important questions, which we will address below.
>
> **W1. Univariate**: Our original focus was on univariate data to align with SOTA methods in TSC XAI [74, 80, 10] . MIX can be straightforwardly applied to multivariate data, as its components are compatible with multi-channel inputs. Concretely, in Phase 1, DWT is applied to each channel independently. Overlapping segments are then generated per channel. Integrated Gradients (IG) & our OSIGV method can be computed for multivariate inputs directly. In Phase 2, the refinement mechanisms also generalize directly. The top-h segments are selected from segments in best view based on their scores. Similarly, the “$\mathrm{KAUC}\tilde{S}_{top}$”​ criterion is calculated by removing k-quantile features based on their timepoint importance scores across channels.
>
> The below Table shows averaged results on UAE Basic Motion on 4 DL models (CNN, PatchTST, Transformer, ResNet34). MIX still outperforms others.
>
> |Metric|InteDisUX|SpectralX|MIX|
> |:---|:---|:---|:---|
> |F@8|0.009 +/- 0.014|0.029 +/- 0.029|**0.060 +/- 0.048**|
> |F@20|0.011 +/- 0.013|0.044 +/- 0.026|**0.106 +/- 0.088**|
> |F@30|0.012 +/- 0.013|0.074 +/- 0.022|**0.157 +/- 0.100**|
> |AUCStop|0.475 +/- 0.094|-|**0.557 +/- 0.093**|
> |F1S|0.208 +/- 0.056|-|**0.299 +/- 0.059**|
> |Robustness|0.398 +/- 0.292|0.381 +/- 0.361|**0.680 +/- 0.217**|
>
> **W2. Computational complexity..**: Please refer to **W3 of Reviewer QTb8**.
>
>
> **Q1. Scores for segments**: While time point-based explanations are common, they can be fragmented and difficult for users to interpret. Providing scores for contiguous segments is often more meaningful because it is better aligned with human perception as pointed out in [80]. For instance, on MIT-BIH ECG, a cardiologist can see that an entire heartbeat segment is responsible for an anomalous prediction. This is significantly more actionable and interpretable than analyzing hundreds of individual, high-importance time points, helping experts to easily validate whether a model's reasoning aligns with their own medical knowledge.
>
> **Q2. Impact of the window sizes**: Our paper included an initial ablation study on the impact of window and step sizes in Figure 4 (G, H, K). We also perform more comprehensive analysis in **W1 of Reviewer QTb8**.
>
> **L1. Validity of the experiments**: The validity of our experiments is grounded in our use of standardized, community-accepted post-hoc evaluation protocols for TSC XAI [10,74,81].
> *For dataset selection*, to ensure a direct and relevant comparison, we use 9 UCR datasets from SpectralX and 2 additional UCR datasets for comprehensive evaluations. *For evaluation metrics*, we adopted the evaluation protocol from SpectralX to ensure fair comparisons. For faithfulness, we follow their method of measuring the drop in model confidence after removing the top-k most important features. For robustness, we also follow their approach of measuring the overlap between the top features of an original instance and a noisy version using Jaccard Coefficient, which is similar to the way SpectralX quantifies the overlap between two fixed-size sets. We also apply $\mathrm{AUC}\tilde{S}_{\text{top}}$ and $\mathrm{F1}\tilde{S}$ from [81].
> Since MIX generates local explanations for each sample, we calculate the faithfulness and robustness scores for every sample in the test set and report averaged values. This is the *standard practice* in post-hoc TSC XAI. We also provide a global explanation in **L6** below.
>
> **L2. Removing segments**: To compare with SpectralX, we need to convert segment-level explanation scores into individual feature-level scores as presented in Section 2.4. The score for each time point $x_j$ is calculated as the maximum absolute score of any segment that contains it. Then we follow the evaluation scheme of SpectralX: measuring Faithfulness@k by removing the top-k most important features and calculating the degradation in the model's confidence score.
>
> To compare with segment-based methods, we use the $AUC\tilde{S}_{\text{top}}$ and $\mathrm{F1}\tilde{S}$​ metrics proposed in [81]. They evaluate performance by measuring the area under the confidence degradation curve as an increasing fraction of the most important features (the top k-quantile) are removed.
>
> Since no evaluation metric is perfect, using these two distinct, standard metrics provides a more robust and trustworthy validation for MIX. To further strengthen our evaluation, we also provide a comparison using the common insertion/deletion metrics in Computer Vision on TwoPatterns dataset in the below Table. MIX outperforms others.
>
> |Metric|InteDisUX|SpectralX|MIX|
> |:---|:---|:---|:---|
> |Insertion|0.6215+/-0.1445|0.8214+/-0.0163|**0.8256+/-0.1433**|
> |Deletion|0.5853+/-0.0107|0.6532+/-0.0324|**0.7086+/-0.0151**|
>
> **L3. How do we know..**: Our method helps users understand which features are influential in a model's decision but does not prove a causal link, which is outside the scope of this study.
>
> Without a perfect causal method, the faithfulness metric is the community's best available proxy for evaluating post-hoc explanations. It directly and pragmatically tests the model's internal behaviour: if the model's confidence drops significantly when we remove the features the explainer identifies as important, it provides strong evidence that the explanation has faithfully captured what the model considers important for its decisions.
> While it cannot reveal the underlying "logic" or "reason" why the model learned to depend on those features, it rigorously validates that our explanation is true to the black-box model's function. This is the standard and accepted goal of attribution-based methods in the current XAI landscape.
>
> **L4. Isn't your approach ..**: For the main results on 11 UCR datasets, we do not average performance across all views. Instead, we report the performance of the single best-performing view for each dataset. Concretely, for each individual view, we average its performance across the different metrics (Faithfulness@8, @20, and @30). We then select the single view with the highest score and report its performance. This process highlights a key strength of our framework: its ability to identify the most effective explanatory perspective for a given task, rather than report the results by averaging. Furthermore, for synthetic and MIT-BIH datasets, we use our greedy selection scheme (Phase 3), which provides an even more direct comparison. In this phase, the most important features are collected from all views and then projected onto a single user-specified view (chosen as $cA_0$). The final explanation's quality is then evaluated on this chosen view only. Therefore, these ensure fair comparisons to single-view methods.
>
> **L5. I found it difficult..**: MIX follows standard practices for evaluating the stability of an explanation by measuring its consistency when the input is slightly perturbed. First, we take the original input time series x and generate an explanation, from which we identify the set of its top-k most important features E. Next, we create a perturbed version of the input x’. By adding a small amount of Gaussian noise. We then generate a new explanation for this noisy input, and identify its corresponding set of top-k features, E’. Finally, we compute the
> Jaccard coefficient between these two sets of top-k features: E, E’. Higher Jaccard score (greater overlap between two sets) signifies that the explanation is stable and consistent. A lower score suggests the explanation is sensitive to minor perturbations and is less robust.
>
> **L6. Local versus global xAI**: Our work follows the post-hoc explanation paradigm. Your suggested work [A] discusses the trade-off in XAI: post-hoc methods risk being unfaithful approximations, while ante-hoc approaches can be less flexible. The paper found that its proposed ante-hoc model can outperform some non-interpretable models for multivariate time series, an interesting result that closes this gap for specific architectures like LSTM & XGBoost.
> While developing effective interpretable-by-design models is a vital direction, we argue that post-hoc explanation remains a crucial and growing area of study. As new increasingly complex architectures like Transformers continue to evolve and achieve SOTA performance on large, complex time series datasets, post-hoc methods remain an essential & practical tool for understanding their behaviour. Moreover, the recent trend towards large, pre-trained foundation models for time series analysis as studied in [B] also strengthen the necessity of post-hoc methods. Even recent hybrid models that combine interpretable and non-interpretable components, such as [87], still require post-hoc analysis to understand their black-box parts.
>
> Regarding the local versus global scope, our original submission focused on local explanations. Inspired by your comment, we have now developed and tested an extension to the MIX framework for generating global explanations. Our approach involves averaging all local attribution maps from all instances to produce a single, summary map. We evaluated its faithfulness and found that it is comparable to local settings, as shown in the table below.
> |Metric|MIX Local|MIX Global|
> |:---|:---|:---|
> |F@8|0.7308+/-0.0230|0.502+/-0.087|
> |F@20|0.7756+/-0.0794|0.693+/-0.016|
> |F@30|0.7438+/-0.0484|0.723+/-0.008|
> |AUCStop|0.707+/-0.015|0.683+/-0.032|
> |F1S|0.347+/-0.024|0.339+/-0.035|
> |Insertion|0.780+/-0.121|0.809+/-0.132|
> |Deletion|0.705+/-0.012|0.687+/-0.028|
>
> Regarding the univariate setting, please refer to **W1** above.
>
> [A] Focuslearn: Fully-interpretable, high-performance modular neural networks for time series, IJCNN, 2024.
> [B] Foundation models for time series analysis: A tutorial and survey, KDD, 2024.

---

> > ### Comment · Reviewer_WeVx · 2025-08-07
> >
> > Thank you for the useful clarifications. Additional results were provided in the rebuttal which are relevant to the questions I had, although I'm not sure these new results can be taken into account at this stage. I accept that the current state of local XAI is as stated, heavily dependent on faithfulness scores and ablation studies, but it's important to acknowledge such limitations (as done in the rebuttal) of XAI approaches that are based on scoring segments. The analysis is interesting but I am not convinced that there is a "most effective explanatory perspective" that can be derived from the available metrics.

---

> ### Author Response · Authors · 2025-08-05
> **Thank you and Follow-up**
>
> Dear Reviewer WeVx,
>
> Thank you again for your very thorough and constructive feedback. We found your suggestions were incredibly valuable in helping us improve our paper. We have posted our rebuttal, which contains several clarifications and new experiments that will be added to our revised version.
>
> In response to your comments, we have now conducted several new experiments. To address weaknesses, a straightforward extension of our method to **multivariate datasets**, and  InteDisUX, SpectralX by flat data to univariate. In addition, we add **computational cost** analysis and measurement. We also answer the question about how meaningful of cores of segment, and answer concern about affection of window size by pointing to analysis on **W1 of Reviewer QTb8**.
> To address your comments on limitation, we propose new **global explanations** by averaging all explanations on TwoPatterns dataset .
>
> We also provided detailed clarifications on several key points: we explained the validity of our experiments by detailing the standard post-hoc evaluation protocols we used; we addressed the question of knowing if "degradation occurs for the right reason"; we confirmed that our comparison is fair as it is based on the single best view, not an average of all views; and we clarified how we measure robustness. Lastly, we included a discussion on local vs. global XAI, informed by the valuable paper you suggested .
>
> We wanted to gently check if these significant new contributions and our other clarifications, all inspired by your review, have helped to address your concerns about the paper's scope.
>
> Thank you for pushing us to improve our work.

---

> ### Author Response · Authors · 2025-08-08
> **Follow-up discussions**
>
> We sincerely thank Reviewer WeVx for acknowledging our rebuttal and for agreeing that the new results we provided are relevant to your original questions. We also appreciate the opportunity to clarify the nature of our new experiments and limitations of our method.
>
>
> *Q1: Thank you for the useful clarifications. Additional results were provided in the rebuttal which are relevant to the questions I had...*
>
>
> We want to emphasize that these new analyses (for multivariate data, global explanations, and insertion/deletion metrics) are not new core technical contributions, but rather demonstrations of the **flexibility and generality of our original MIX framework**. Our core method was designed to be adaptable from the start. For instance, applying MIX to multivariate data is a straightforward extension, as the DWT and IG-based components naturally operate on a per-channel basis. Similarly, our new global explanation is an aggregation of the local explanations, which was already designed to produce by our framework, and the insertion/deletion metrics are simply an additional, standard way to evaluate these same explanations. *Therefore, while we kept our original submission focused on univariate data to align with SOTA baselines, we hope these new results are seen as a validation of our framework's robust and flexible design*. Indeed, we are grateful that your insightful questions gave us the opportunity to demonstrate this. We are going to add these new results in our revised version.
>
>
> *Q2: I accept that the current state of local XAI is as stated, heavily dependent on faithfulness scores and ablation studies, but it's important to acknowledge such limitations (as done in the rebuttal) of XAI approaches that are based on scoring segments...*
>
>
> We sincerely thank the reviewer for this deep and insightful point, and **we agree completely**. You have raised a crucial challenge that is at the heart of XAI research: that **there is no single, perfect metric to determine a "most effective explanatory perspective."** And **segment-based methods (including our MIX) are also not an exception**. This issue extends across the entire field; for instance, there are no universal metrics to definitively decide between ante-hoc vs. post-hoc approaches, global vs. local scopes, **segment vs. time point** in TSC XAI, or even between different explanation types like prototypes and attribution maps. This is also a key takeaway from recent studies like Retzlaff et al. [A], which propose complex decision trees (Figure 1 of [A]) precisely because **there is no single best answer, and the optimal choice always depends on the specific context and trade-offs of the project**.
>
>
> Though our approach still suffers from that problem, given this landscape, our work takes a pragmatic and principled approach. As you noted, we rely on the current community standards for **local post-hoc XAI**, which our work focuses on: faithfulness scores and ablation studies. To select the best view for our main comparisons against SOTA methods, we used a composite criterion that averages multiple standard faithfulness scores (F@8, F@20, and F@30). *We believe this is a robust and transparent way to make a selection based on the best-available tools the field currently offers*. For fair comparison, we use different metrics with other different aspects including robustness, $AUC\tilde{S}_{\text{top}}$ and $\mathrm{F1}\tilde{S}$ to compare with SOTA.
>
>
> Besides, **our MIX framework was designed with an assumption about “no perfect metric” in mind**. Hence it is not rigidly tied to a specific metric. The framework's key strength is its flexibility. The selection criterion used to identify the "best view" is modular. As the XAI field matures and develops better and more holistic evaluation metrics, they can be easily integrated into our framework to replace the current faithfulness average. The adaptive selector in Phase 2, which acts as a quality gate, is a key example of this modular and principled design. In addition, we can combine different metrics to have a better evaluation (by covering wider aspects) like the way we did in our paper with 3 metrics. This is inspired from the way different classification evaluation metrics e.g. Acc, F1, AUC are combined in ML classification problems.
>
>
> In summary, **we fully agree with your point on the limitations of current metrics and segment-based approaches including MIX**. We are very grateful for your feedback, which has pushed us to articulate this important aspect of our design more clearly. **We are going to acknowledge the limitations of segment-based XAI approaches clearly in the limitation part as suggested.**
>
>
> Thanks again, If there is any further clarification that would be helpful for your final evaluation, we would be more than happy to provide it.
>
>
> References:
>
>
> [A] Post-hoc vs ante-hoc explanations: xAI design guidelines for data scientists, 2024.

---

### Official Review · Reviewer_4jMc · 2025-07-03

**Clarity:** 2
**Significance:** 2
**Originality:** 2
**Rating:** 3
**Confidence:** 3

**Summary:**

The paper proposes MIX, a multi-view explanation framework for time series classification that leverages Haar Discrete Wavelet Transform (DWT) to create multiple time-frequency views. The framework includes three phases: independent explanation generation for each view, cross-view refinement, and greedy feature selection. The authors show improvements over existing methods including LIMESegment, InteDisUX, and SpectralX.

**Questions:**

The "best view" is selected using KAUCS, which depends on the explanation quality. But then this view is used to refine other views' explanations. Is it a chicken-and-egg problem?

**Ethical Concerns:**

["NO or VERY MINOR ethics concerns only"]

**Final Justification:**

I will maintain the score

**Limitations:**

Yes

**Quality:**

2

**Strengths And Weaknesses:**

**Strengths**
- Multi-view perspective: The idea of using multiple wavelet decomposition levels to create different views is interesting and could capture patterns at different resolutions.
- Comprehensive evaluation: The paper includes extensive experiments across 11 UCR datasets with 3 different architectures.

**Weaknesses**
- Not the First Multi-view/Time-Frequency Work. As noted, this is not the first work in time-frequency explanation. SpectralX (cited as [10]) already proposed time-frequency based explanations using STFT. The authors acknowledge this but claim their approach is better because: SpectralX uses "fixed STFT setup" while MIX uses adaptive Haar DWT and SpectralX has "randomness issues". However, these differentiations feel incremental rather than fundamentally novel.
- Experiments: It didn't compare with SpectralX, which is frequently referred by the authors, by saying "is not included to comparison as it
cannot generate explanations in the raw time domain from spectrogram". However, SpextralX is able to present faithfulness results with standard deviations in both time and time-frequency domains. It's easy and necessary to implement similar experiments.
- Overly Complex and Heuristic Design. The framework is extremely complex with many heuristic choices Figure 1 is indeed overwhelming: It tries to show all three phases, multiple attribution mechanisms (IGV, OSIGV, KIGV), and the cross-view refinement process. This complexity makes it difficult to understand the core contribution.


**Summary**
While the paper tackles an important problem and shows empirical improvements, the contribution is incremental given existing work like SpectralX. The framework is overly complex with many heuristic design choices that lack theoretical justification.

---

> ### Author Rebuttal · Authors · 2025-07-30
>
> We sincerely thank Reviewer 4jMc for the detailed and constructive feedback. We are grateful that you found our multi-view perspective "interesting" and our evaluation "comprehensive". We would like to clarify your main concerns below.
>
> **W1. Not the First...** We agree that our method is not the first to utilize the time-frequency space for explaining time series classification (TSC). Hence, our introduction clarified that *“we propose a new perspective for TSC XAI that explains models from multiple views of time series in both time-frequency domains”*.
> However, MIX utilizes an independent interactive multiple time-frequency view explanation scheme, in contrast to the single view explanation scheme used by SpectralX and others. Views in MIX are created via Discrete Wavelet Transform (DWT), thus forming a hierarchical structure at multiple time-frequency resolutions. This hierarchical relationship not only can be exploited to provide explanations from multiple time-frequency perspectives but also can be used to enhance explanations of all views by letting them interactively refine each other via a proposed process named *across-view refinement strategy*. SpectralX uses STFT to create a spectrogram and explain on it. Though it is also a time-frequency based method like MIX, it does not provide multiview explanation results. Overall, this *interactive multi-view explanation scheme* is the most unique property of MIX and is the key difference between MIX and all existing works (including SpectralX) to the best of our knowledge.
> Furthermore, MIX is the first method that introduces a novel attribution approach of applying Integrated Gradients (IG) to overlapping segments in the wavelet space, which distinguishes it from prior segment-based methods such as InteDisUX and LIMESegment.
> Finally, our attribution mechanism OSIGV is fundamentally distinct from that of SpectralX and others. We also propose a way to resolve issues of instability and potential false negatives (c.f. Theorem B.1 in Appendices) that can arise from SpectralX's sampling-based scheme. This exploration not only justifies our design choice but also provides a new insight and awareness for the XAI community on the importance of stable attribution.
> The below Table summarises key differences between MIX and SpectralX.
>
> |Aspects|SpectralX|MIX|
> |:---|:---|:---|
> |Multi-views|No|Yes|
> |Time-frequency transform|STFT|DWT|
> |View interaction|No|Yes|
> |Attribution mechanism|Insertion/Deletion| OSIGV (solving stability and false negative issues)|
>
> In summary, MIX is different to SpectralX and others in 3 fundamental aspects: (i) hierarchical *multi-view explanations* in multi-resolution time-frequency domains, (ii)  *interactive explanation refinement scheme among views*, and (iii) novel *attribution mechanism*.
>
> **W2. Experiments**: We apologize if the location of this analysis was not prominent in our submission.
>
> We would like to respectfully guide the reviewer to *Figure 2* and *Section 3.2 (Main results & Results on UCR datasets)* in the paper, where we compare MIX directly with other methods, including *SpectralX*, LIMESegment and InteDisUX, across our full set of 11 UCR datasets and 3 deep learning architectures. The results show that MIX achieves SOTA performance in 24/33 cases for faithfulness and 27/33 cases for robustness.
>
> To provide a more granular view of the results in Figure 2, the table below details the mean and standard deviation for each dataset for Faithfulness@30 (F@30), averaged over the three deep learning architectures for every dataset.
>
> |Dataset|MIX|SpectralX|
> |:---|:---|:---|
> |ArrowHead|0.394 +/- 0.357|**0.6112 +/- 0.3778**|
> |Strawberry|0.1951 +/- 0.1913|**0.5638 +/- 0.3732**|
> |Yoga|0.3891 +/- 0.1280|**0.4486 +/- 0.0266**|
> |FordA|**0.3254 +/- 0.1382**|0.0813 +/- 0.0163|
> |FordB|**0.3128 +/- 0.1401**|0.1463 +/- 0.0633|
> |MixedShapesRegularTrain|**0.5808 +/- 0.2114**|0.5036 +/- 0.1730|
> |CinCECGTorso|**0.4849 +/- 0.2829**|0.3850 +/- 0.2986|
> |GPGender|0.5003 +/- 0.0345|**0.5190 +/- 0.0607**|
> |TwoPatterns|**0.7438 +/- 0.0484**|0.6266 +/- 0.0748|
> |MixedShapesSmallTrain|**0.6710 +/- 0.0826**|0.6036 +/- 0.0826|
> |Wafer|**0.6538 +/- 0.1417**|0.1585 +/- 0.0954|
>
> The above results show a clear advantage for MIX compared to SpectralX. Across the 11 UCR datasets, MIX outperforms SpectralX in terms of Faithfulness@8 on 9 datasets, Faithfulness@20 on 8 datasets, and Faithfulness@30 on 7 datasets. In terms of robustness, MIX is better in all 11 datasets. This strong overall performance is also reflected in the grouped box plot with standard deviation in Figure 2, which shows that MIX has the best average scores across over all 33 experimental setups for all four metrics.
>
> The below Table also shows that MIX outperforms SpectralX in terms of faithfulness and robustness for MIT-BITH data.
>
>
> |Metric|IntedisUX|SpectralX|MIX|
> |:---|:---|:---|:---|
> |F@30|0.6656|0.7478|0.8371|
>
>
> **W3. Overly Complex..**: Our framework has several components and Figure 1, in its attempt to be comprehensive, can appear overwhelming. We agree that our explanation of the method's design can be improved, and we will revise both the text and Figure 1 to provide more intuitive descriptions. Our framework is built systematically from well-established principles, where each new component serves a specific purpose to address limitations in prior work. Please find below step-by-step  justifications of our designs (IGV, OSIGV, KIGV, cross-view refinement).
>
> *For IGV*: our method is built upon Integrated Gradients (IG) , a widely used and axiomatically sound attribution method. IGV, is the necessary mathematical adaptation of IG to operate in a transformed "view" space (e.g., the wavelet domain). As defined in our paper, if a view transformation V is differentiable and invertible, IGV is simply IG applied to the composite model $F \circ V^{-1}$. This provides a mathematically sound way to calculate feature importance in any given view, not just the input space.
>
> *For OSIGV*: to make the point-wise scores from IGV more aligned with human perception, we aggregate them into segments. Our method OSIGV (Overlapping Segment-level Integrated Gradients for a View) is designed to improve upon prior segment-based approaches and make explanations more robust and interpretable. First, we use IGV to calculate an importance score for each individual point in a given view V(x). Second, we generate a set of overlapping segments across the view, where each segment is defined by a window size. The final score for each segment is then calculated as the sum of the IGV scores of all the individual points contained within that segment, as defined in our paper. The use of overlapping segments is a key design choice that directly addresses a limitation in prior work like InteDisUX, which uses non-overlapping segments. Non-overlapping segments risk splitting meaningful temporal patterns and can miss important information that occurs at the boundaries between segments.
>
> *For Interactive Refinement (KIGV & Cross-View Refinement)*: the perceived complexity of our framework arises from our main contribution: cross-view refinement, which is enabled by our proposed Keystone-first Integrated Gradients (KIGV) method. Our motivation comes from the "keystone species" concept in ecology; just as a keystone species has a disproportionate impact on its environment, we hypothesize that certain "keystone features" in one view are strongly indicative of the model's core reasoning . Our novel KIGV method leverages this idea by first attributing importance to these keystone features, which are approximated as the top-h segments from the best view identified in Phase 1. By focusing the IG calculation path through these important regions first, we can generate a more robust gradient signal with less noise, and this refined attribution map is then used to improve the explanations in other views. To ensure that the cross-view refinement step is always beneficial, we introduce a final safeguard: the adaptive selector. The refined explanation generated by KIGV is only adopted for a given view if it shows a demonstrable improvement over the original one from Phase 1. This ensures our "heuristic" choice is not arbitrary but is instead a principled decision controlled by an objective quality metric based on paper [81].
>
> **Q1. Chicken-and-egg..**: We are grateful to the reviewer for this insightful question, as it addresses a crucial point about the framework's logical flow. There is no "chicken-and-egg" problem because the process is strictly sequential, which avoids any circular dependency.
>
> First, in *Phase 1*, the framework generates an initial, independent explanation for every view using the OSIGV attribution mechanism. This step is foundational, creating a complete set of unrefined explanations. At this stage, each explanation is generated in isolation based solely on its own view, and no refinement or "best view" selection has yet occurred. This provides the baseline set of explanations upon which the next phase will operate.
>
> MIX then moves to *Phase 2*, which is a two-step procedure that clearly separates selection from refinement. In the first step, the quality of the initial explanations from Phase 1 is evaluated using the $KAUC\tilde{S}_{top}$ score. The view that achieves the highest score is designated as the "best view" ($V$). Only *after* this selection is complete, does the refinement process begin. The information from the now-selected $V$ is used as a static guide to refine the explanations of the other views using the Keystone-first IG (KIGV) method.
>
> In summary, the selection of the best view depends on the quality of the initial, unrefined explanations, and the refinement process depends on the result of that selection. The refined explanations do not influence the initial scores that were used to make the selection in the first place. This strictly linear process ensures that there is no circular logic.

---

> > ### Comment · Reviewer_4jMc · 2025-08-06
> >
> > Thank authors for the detailed rebuttal and effort. I appreciate your clarifications. However, I respectfully maintain my original score.
> >
> > In particular, the concern regarding the complexity of the method design have not been sufficiently addressed. This aspect remains a significant issue affecting my evaluation.

---

> ### Author Response · Authors · 2025-08-05
> **Thank you and Follow-up**
>
> Dear  Reviewer 4jMC,
>
> Thank you again for your detailed and constructive review. We have posted our rebuttal and wanted to follow up on your key suggestions.
>
> In our response, we worked to clarify our core novelty in the multi-view setup and the interactive refinement scheme. Moreover, in response to your feedback, we pointed to our existing comparisons with SpectralX in Figure 2 and Section 3.2 and conducted a new faithfulness comparison on the expert-annotated MIT-BIH dataset. Lastly, we addressed your question regarding the "chicken-and-egg" problem.
>
> We were hoping to check if this new evidence and our other clarifications help to address your main concerns.
>
> Thank you for your time and guidance.

---

> ### Author Response · Authors · 2025-08-06
> **Thank you and Complexity Discussion**
>
> Thank you, Reviewer 4jMc, for the follow-up comment and for appreciating our detailed rebuttal. We are very grateful that you have narrowed down the discussion to **your specific remaining concern about the complexity of the method's design**. This allows us to provide a more focused final clarification on this important point.
> We agree that our framework has several components, and **we hope this clarification shows that our algorithm design is not arbitrarily complex**. Instead, each piece is a necessary and justified step to overcome limitations in prior work. In your initial review, you correctly identified our framework's main components: its three attribution mechanisms (IGV, OSIGV, KIGV) and the cross-view refinement process. **You noted that, due to these components, "This complexity makes it difficult to understand the core contribution"**. We hope to use this response to clarify how each part provides a necessary and justified solution to **the challenges of our new proposed multi-view explanation concept**.
>
> Our three-phase design is a direct and logical workflow for **tackling the novel challenge of creating a very novel interactive multi-view explanation framework**. Because **no prior work has addressed this specific problem** (to the best of our knowledge), a systematic, multi-step process was required to explore each view, connect the explanations between them, and consolidate the final insights. The role of each phase is distinct and necessary. **Phase 1** is the logical first step: generating an independent explanation for each view to establish a baseline. **Phase 2** performs our core novel contribution: the cross-view refinement where the "best view" is used to improve the others. **Phase 3** addresses a crucial usability need by consolidating these multi-view insights into a single, user-focused explanation. We acknowledge that designing a comprehensive flow for this novel multi-view setting was a significant effort. We were encouraged that other reviewers, such as Reviewer QTb8, recognized this systematic and interactive approach as a key strength of our work.
>
> Each of our attribution mechanisms was designed **not as an arbitrary heuristic**, but as a direct solution to a specific and important challenge in the new proposed Multi-view perspective for XAI. **IGV** is not a heuristic choice but is the fundamental, mathematically sound formula for applying the axiomatic IG method in a transformed view space (c.f. Definition 2.4 in the main paper). As clarified in our rebuttal, this provides a necessary and principled way to calculate feature importance in any given view. **OSIGV** was designed to solve a central problem in XAI: human perception alignment. By aggregating scores into overlapping segments, it produces more robust and interpretable explanations than the fragmented, point-wise scores from IG or the non-overlapping segments of prior work . This improved robustness also better aligns with human reasoning, as a person's explanation is typically stable against minor, irrelevant noise, leading to a high robustness score as in an ablation study below in **Tables 1 and 2 of W2 of the reviewer QTb8** and below. **No previous work deals with overlapped segments in TSC XAI**. **KIGV** is the novel mechanism that enables our paper's core contribution: cross-view refinement. As detailed in the rebuttal, this method uses "keystone" features from the most faithful view to create a less noisy gradient signal when explaining other views . This focus on the most important features is the key to improving the faithfulness of each view's explanation, as empirically demonstrated in our paper's ablation studies. In addition, to ensure this powerful refinement is not just a blind heuristic, we added the **adaptive selector**. This component acts as a principled quality gate, only accepting a refined explanation if it's a proven improvement according to our objective quality metric. This technique helps ensure our algorithm consistently provides a more faithful explanation.
>
> *In summary, **MIX's design is not complex for its own sake, but is a direct result of our attempt to solve several key challenges of this novel multi-view XAI setup at once**. To create a robust multi-view explanation system, our framework was built to simultaneously address: providing an axiomatically-grounded explanation for any given view space (handled in Phase 1), Enabling views to interact and demonstrably improve one another's faithfulness (the core of Phase 2), Ensuring the final, user-facing explanation is of the highest possible quality (the goal of the Adaptive Selector in Phase 2 and the Greedy Selection in Phase 3).*
>
> We provide ablation study in our submission to clarify contributions of each component for its aspect in the next comment below.
>
> **(To be continued…)**

---

> ### Author Response · Authors · 2025-08-06
> **Ablation Studies for Core Contribution of each component**
>
> We would also like to **respectfully point to the theoretical and empirical evidence within our submission that justifies each of our design choices, demonstrating that they are good attempt to deal with mult-view explanation**.
>
> The foundation of our approach, IGV, is justified by its **mathematical principles as a formal definition of Integrated Gradients for a transformed view space**, which we present in Definition 2.4 of our paper.
>
> The value of OSIGV's segment-based approach is supported by sensitivity analysis displayed in below Tables 1 and 2, which show that robustness tends to increase with a larger window size, indicating that OSIGV can help generate more robust and human-aligned explanations. This shows how the choice of segment size directly impacts human-aligned. We also provide visualizations to show **how overlapping segments can appear as a human-aligned explanation** in Fig 4,5,6,7,8,9,10,11, 12 in Supplementary Material (this is different from Appendices concatenated to the main paper).
>
> _Table 1: Window/Step on FordA_
> |Window Size|Step Size|F@30|Robustness|
> |:---|:---|:---|:---|
> |8|4|0.3238+/-0.1445|0.6527+/-0.1054|
> |16|8|0.2480+/-0.0661|0.6647+/-0.1350|
> |32|16|0.2111+/-0.0749|0.6732+/-0.1449|
> |48|24|0.2014+/-0.0829|0.7379+/-0.0560|
> |56|28|0.1967+/-0.0884|0.7172+/-0.0446|
> |64|32|0.1789+/-0.0761|0.6937+/-0.0480|
> |96|48|0.2116+/-0.0885|0.7501+/-0.0710|
> |128|64|0.2153+/-0.0857|0.7614+/-0.0676|
>
>
> _Table_2: Window/Step on TwoPatterns_
> |Window Size|Step Size|F@30|Robustness|
> |:---|:---|:---|:---|
> |8|4|0.7438+/-0.0484|0.8608+/-0.0532|
> |16|8|0.7508+/-0.0587|0.8739+/-0.1043|
> |32|16|0.7224+/-0.0130|0.9337+/-0.0349|
> |48|24|0.7150+/-0.0200|0.9716+/-0.0253|
> |56|28|0.7214+/-0.0123|0.9907+/-0.0161|
> |64|32|0.7150+/-0.0200|0.9934+/-0.0114|
>
> Crucially, the benefits of our novel components are demonstrated in our ablation studies. The significant value of our cross-view refinement (Phase 2), which is enabled by KIGV and the adaptive selector, is empirically proven in Figure 4 (A, B, C). These plots clearly show that faithfulness consistently and significantly increases across all views after the refinement process is applied. To further isolate the contribution of the  **adaptive selector**, we added a new ablation study in Fig 3 in Supplementary Material, which shows its direct impact on improving faithfulness.
>
> Finally, the contribution of the greedy selection (Phase 3) is quantified in Table 1 of our main paper. The results clearly show that the "w/ Greedy" version of our method achieves a higher Jaccard score on both the Synthetic and MIT-BIH ground-truth datasets, demonstrating its effectiveness in improving the quality of the final, user-focused explanation .
>
> **We hope these clarifications through the existing evidence in our paper helps to demonstrate that our design choices are both principled and empirically validated.**

---

### Comment · Area_Chair_7c2W · 2025-08-03
**Author–Reviewer Discussion: Early Participation Appreciated**

Dear Reviewers,

Please take time to read the other reviews and author responses carefully, and actively participate in the Author–Reviewer Discussions—posting an initial comment early, even a brief one, helps enable a constructive exchange. Thank you!

---

### Author Response · Authors · 2025-08-09
**Thank You to the Reviewers and Area Chair**

Dear Area Chair and Reviewers,

As the author-reviewer discussion period concludes, we wanted to sincerely thank you all for your time and for a thorough and constructive review process. We would also like to extend a special thank you to our Area Chair for helping to facilitate a productive discussion.

Your collective feedback has been invaluable and has directly led to what we believe are significant improvements to our paper, including new experiments and important clarifications.

We hope that our detailed rebuttal and these new results provide a clear basis for the next stage of evaluation. We are grateful for the opportunity to improve our work based on your guidance.

Thank you again.

Sincerely,

The Authors

---

### Note · Authors · 2025-08-12

We sincerely thank all reviewers and our Area Chair for a thorough, constructive, and highly valuable review process.

We were particularly encouraged by the consensus among the reviewers on our paper's core strengths. There was broad agreement on the **novelty of our interactive multi-view perspective** and the **comprehensiveness of our experimental setups and baseline comparisons**.

The discussion period was also extremely productive. We were very pleased that our detailed rebuttal and the significant new work we performed were able to fully resolve the concerns of Reviewer QTb8 and Reviewer BmjP, who both kindly posted final comments confirming their satisfaction with our responses.

Furthermore, our discussions with Reviewers 4jMc and WeVx were particularly beneficial. We are grateful that after our detailed rebuttal progressed from multiple initial concerns to more focused final points. Reviewer 4jMc, for instance, narrowed their initial concerns on novelty and experimental comparisons down to a final, focused question on our framework's complexity. Similarly, the discussion with Reviewer WeVx evolved from several initial weaknesses and questions (regarding the univariate setup, computational complexity, and validity of evaluation) to a thoughtful, high-level exchange on the inherent challenges of XAI evaluation and the necessity of acknowledgement of limitation of segment-based approaches including our MIX method, InteDisUX, and LIMESegment.

We provided detailed final responses to address these last focused points, and while the discussion period concluded before we could receive their final thoughts, we hope these clarifications provide sufficient information for their final evaluation. We truly appreciate their deep engagement, which gave us the opportunity to strengthen our paper by resolving the key concerns they raised.

As a direct result of this excellent feedback, our paper has been substantially improved with new experiments demonstrating **multivariate and global explanation capabilities**, **new sensitivity and complexity analyses**, and **clearer justifications for our design choices**. We are confident that the final paper is a much stronger and more complete contribution thanks to this process.

Thank you again for your time and consideration.

---

### Decision · Program_Chairs · 2025-09-17

**Decision:**

Accept (poster)

**Comment:**

This paper presents MIX, a multi-view explanation framework for time series classification, combining Haar DWT-based multi-resolution views, cross-view refinement, and novel attribution mechanisms (IGV, OSIGV, KIGV). Reviewers noted that the paper is well-motivated, technically coherent, and demonstrates good empirical performance across multiple datasets and architectures. While some concerns remain regarding framework complexity, limited quantitative evaluation, and sensitivity to hyperparameters, these do not entirely diminish the potential of the approach. Overall, the work provides an interesting direction for improving time series interpretability.